# Learning to Discover Skills through Guidance

**Hyunseung Kim**[*,1]  **Byungkun Lee**[*,1]  **Hojoon Lee**[1]
**Dongyoon Hwang**[1]  **Sejik Park**[1]  **Kyushik Min**[2]  **Jaegul Choo**[1]
[1]Kim Jaechul Graduate School of AI, KAIST.    [2]KAKAO Corp.
{mynsng, byungkun.lee, joonleesky,
godnpeter, sejik.park, jchoo}@kaist.ac.kr
queue.min@kakaocorp.com

## Abstract

In the field of unsupervised skill discovery (USD), a major challenge is limited exploration, primarily due to substantial penalties when skills deviate from their initial trajectories. To enhance exploration, recent methodologies employ auxiliary rewards to maximize the epistemic uncertainty or entropy of states. However, we have identified that the effectiveness of these rewards declines as the environmental complexity rises. Therefore, we present a novel USD algorithm, skill **disco**very with gui**dance** (**DISCO-DANCE**), which (1) selects the *guide skill* that possesses the highest potential to reach unexplored states, (2) guides other skills to follow *guide skill*, then (3) the guided skills are dispersed to maximize their discriminability in unexplored states. Empirical evaluation demonstrates that DISCO-DANCE outperforms other USD baselines in challenging environments, including two navigation benchmarks and a continuous control benchmark. Qualitative visualizations and code of DISCO-DANCE are available at https://mynsng.github.io/discodance/.

## 1   Introduction

Deep Reinforcement Learning (DRL) has shown remarkable success in a wide range of complex tasks, from playing video games [28, 49] to complex robotic manipulation [4, 14]. However, the majority of DRL models are designed to train from scratch for each different task, resulting in significant inefficiencies. Furthermore, reward functions adopted for training the agents are generally handcrafted, acting as an impediment that prevents DRL from scaling for real-world tasks. For these reasons, there has been an increasing interest in training task-agnostic policies without access to a pre-defined reward function. One approach that has been widely studied for achieving this is unsupervised skill discovery (USD), which is a training paradigm that aims to acquire diverse and discriminable behaviors, referred to as skills [7, 11, 13, 17, 22, 25, 47, 52, 32, 20, 46]. These pre-trained skills can be utilized as useful primitives or directly employed to solve various downstream tasks.

Most of the previous studies in USD discover a set of diverse and discriminable skills by maximizing the self-supervised, intrinsic motivation as a form of reward [1, 11, 13, 17, 47]. Commonly, mutual information (MI) between the skill's latent variables and the states reached by each skill is utilized as the self-supervised reward. However, it has been shown in recent research that solely maximizing the sum of MI rewards is insufficient in exploring the state space because, asymptotically, the agent receives larger rewards for visiting known states rather than for exploring novel states due to the high MI reward it receives at a fully discriminable state [7, 25, 52, 20].

To ameliorate this issue, recent studies designed an auxiliary exploration reward that incentivizes the agent when it succeeds in visiting novel states [52, 24, 25]. However, albeit provided these auxiliary

---

[*]Equal contributions.

37th Conference on Neural Information Processing Systems (NeurIPS 2023).

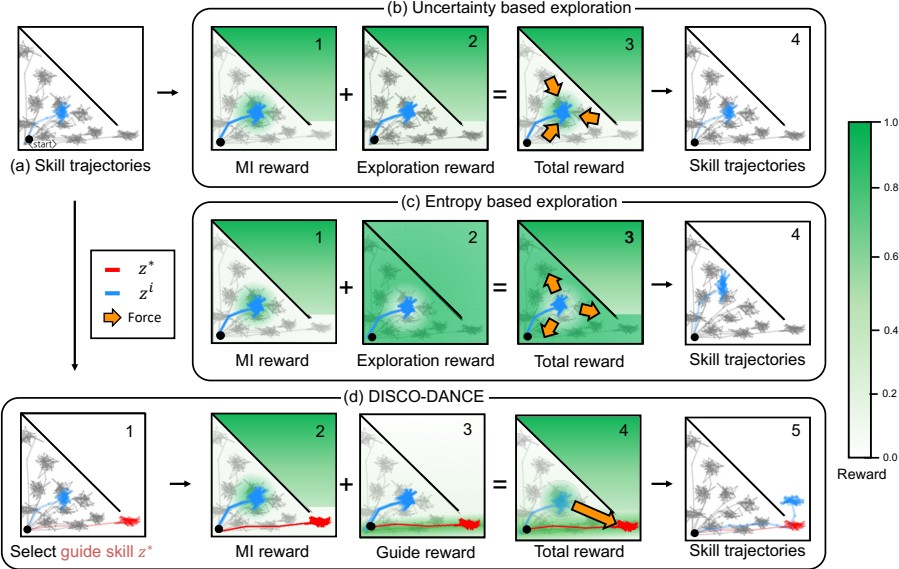

Figure 1: **Conceptual illustration of previous methods and DISCO-DANCE.** Each skill is shown with a grey-colored trajectory. Blue skill $z^i$ indicates an unconverged skill, and the reward landscape of $z^i$ is represented in green. Here, (b,c) illustrates a reward landscape of previous methods, DISDAIN, APS, and SMM. (b) DISDAIN fails to reach the upper region due to the absence of a pathway to the unexplored states. (c) APS and SMM fail since they do not provide exact directions to the unexplored states. On the other hand, (d), DISCO-DANCE directly guides $z^i$ towards selected guide skill $z^*$ which has the highest potential to reach the unexplored states. A detailed explanation of the limitations of each baseline is described in Section 2.2.

rewards, previous approaches may exhibit decreased effectiveness in challenging environments. Fig. 1 conceptually illustrates the ineffectiveness of previous methods. Suppose that the upper region of Fig. 1a is difficult to reach with MI rewards, resulting in obtaining skills that are stuck in the lower-left region. To make these skills explore the upper region, previous methods provide auxiliary exploration rewards using intrinsic motivation (e.g., disagreement, curiosity-based bonus). However, since they do not indicate exactly which *direction* to explore, it becomes more inefficient in challenging environments. We detail the limitations of previous approaches in Section 2.2.

In response, we propose a new exploration objective that aims to provide *direct guidance* to the unexplored states. To encourage skills to explore unvisited states, we first identify a guide skill $z^*$ which possesses the highest potential for reaching unexplored states (Fig. 1d-1). Next, we select skills that are relatively unconverged, and incentivize them to follow the guide skill $z^*$ in an effort to leap over the state regions with low MI rewards (Fig. 1d-2:4). Finally, they are dispersed to maximize their distinguishability (Fig. 1d-5), resulting in obtaining a set of skills with high state coverage. We call this algorithm as skill **disco**very with gui**dance (DISCO-DANCE)** and is further presented in Section 3. DISCO-DANCE can be thought of as filling the pathway to the unexplored region with a positive dense reward.

Through empirical experiments, we demonstrate that DISCO-DANCE outperforms previous approaches in terms of state space coverage and downstream task performances in two navigation environments (2D mazes and Ant mazes), which have been commonly used to validate the performance of the USD agent [7, 19]. Furthermore, we also experiment in Deepmind Control Suite [53], and show that the learned set of skills from DISCO-DANCE provides better primitives for learning general behavior (e.g., run, jump, and flip) compared to previous baselines.

## 2 Preliminaries

In Section 2.1, we formalize USD and explain the inherent pessimism that arises in USD. Section 2.2 describes existing exploration objectives for USD and the pitfalls of these exploration objectives.

## 2.1 Unsupervised Skill Discovery and Inherent Exploration Problem

Unsupervised Skill Discovery (USD) aims to learn *skills* that can be further utilized as useful primitives or directly used to solve various downstream tasks. We cast the USD problem as discounted, finite horizon Markov decision process $\mathcal{M}$ with states $s \in \mathcal{S}$, action $a \in \mathcal{A}$, transition dynamics $p \in \mathcal{T}$, and discount factor $\gamma$. Since USD trains the RL agents to learn diverse skills in an unsupervised manner, we assume that the reward given from the environment is fixed to 0. The skill is commonly formulated by introducing a skill's latent variable $z \in \mathcal{Z}$ to a policy $\pi$ resulting in a latent-conditioned policy $\pi(a|s, z)$. Here, the skill's latent variable $z$ can be represented as a one-hot vector (i.e., discrete skill) or a continuous vector (i.e., continuous skill). In order to discover a set of diverse and discriminable skills, a standard practice is to maximize the mutual information (MI) between state and skill's latent variable $I(S; Z)$ [1, 11, 13, 17, 47].

$$I(Z, S) = -H(Z|S) + H(Z) = \mathbb{E}_{z \sim p(z), s \sim \pi(z)}[\log p(z|s) - \log p(z)] \tag{1}$$

Since directly computing the posterior $p(z|s)$ is intractable, a learned parametric model $q_\phi(z|s)$, which we call *discriminator*, is introduced to derive lower-bound of the MI instead.

$$I(Z, S) \geq \tilde{I}(Z, S) = \mathbb{E}_{z \sim p(z), s \sim \pi(z)}[\log q_\phi(z|s) - \log p(z)] \tag{2}$$

Then, the lower bound is maximized by optimizing the skill policy $\pi(a|s, z)$ via any RL algorithm with reward $\log q_\phi(z|s) - \log p(z)$ (referred to as $r_{\text{skill}}$). Note that each skill-conditioned policy gets a different reward for visiting the same state (i.e., $r_{\text{skill}}(z_i, s) \neq r_{\text{skill}}(z_j, s)$). It results in learning skills that visit different states, making them discriminable.

However, maximizing the MI objective is insufficient to fully explore the environment due to *inherent pessimism* of its objective [7, 25, 52]. When the discriminator $q_\phi(z|s)$ succeeds in distinguishing the skills, the agent receives larger rewards for visiting known states rather than for exploring novel states. This lowers the state coverage of a given environment, suggesting that there are limitations in the set of skills learned (i.e., achieving a set of skills that only reach a limited state space).

## 2.2 Previous exploration bonus and Its limitations

In order to overcome the *inherent pessimistic exploration* of USD, recent studies have attempted to provide auxiliary rewards. DISDAIN [52] trains an ensemble of $N$ discriminators and rewards the agent for their disagreement, represented as $H(\frac{1}{N} \sum_{i=1}^{N} q_{\phi_i}(Z|s)) - \frac{1}{N} \sum_{i=1}^{N} H(q_{\phi_i}(Z|s))$. Since states that have not been visited frequently will have high disagreement among discriminators, DISDAIN implicitly encourages the agent to move to novel states. However, since such exploration bonus is a *consumable* resource that diminishes as training progresses, most skills will not benefit from this if other skills reach these new states first and exhaust the bonus reward.

We illustrate this problem in Fig. 1b. Suppose that all skills remain in the lower left states, which are easy to reach with MI rewards. Since the states in the lower left region are accumulated in the replay buffer, disagreement between discriminators remains low (e.g., low exploration reward in the lower left region). Therefore, there will be no exploration reward left in these states. This impedes $z_i$ from escaping its current state to unexplored states, as shown in Fig. 1b-4.

On the other hand, SMM encourages the agent to visit states where it has not been before using a learned density model $d_\theta$ [24]. APS incentivizes the agent to maximize the marginal state entropy via maximizing the distance of the encoded states $f_\theta(s_t)$ between its k nearest neighbor $f_\theta^k(s_t)$ [25].

$$\begin{aligned} r_{\text{exploration}}^{\text{SMM}} &\propto -\log d_\theta(s) \\ r_{\text{exploration}}^{\text{APS}} &\propto \log ||f_\theta(s) - f_\theta^k(s)|| \end{aligned} \tag{3}$$

These rewards push each skill out of its converged states (in Fig. 1c). However, they still do not provide a *specific direction* on where to move in order to reach unexplored states. Therefore, in a difficult environment with a larger state space, it is known that these exploration rewards can operate inefficiently [7, 9]. In the next section, we introduce a new exploration objective for USD which addresses these limitations and outperforms prior methods on challenging environments.

## 3 Method

Unlike previous approaches, we design a new exploration objective where the *guide skill* $z^*$ directly influences other skills to reach explored regions. DISCO-DANCE consists of two stages: (i) selecting

guide skill $z^*$ and the *apprentice skills* which will be guided by $z^*$, and (ii) providing guidance to apprentice skills via *guide reward*, which will be described in Section 3.1 and 3.2, respectively.

## 3.1 Selecting Guide skill and Apprentice Skills

**Guide skill.** We recall that our main objective is to obtain a set of skills that provides high state space coverage. To make other skills reach the unexplored state, we define the *guide skill* $z^*$ as the skill which is most *adjacent* to the unexplored states. One naive approach in selecting the guide skill is to choose the skill whose terminal state is most distant from the other skills' terminal state (e.g., blue skill in Fig. 2b). However, such a selection process does not take into account whether the guide skill's terminal state is neighboring promising *unexplored states*. In order to approximate whether a skill's terminal

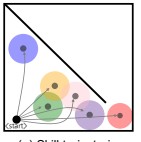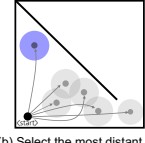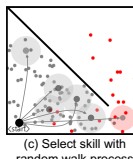

(a) Skill trajectories  (b) Select the most distant skill  (c) Select skill with random walk process

Figure 2: **Guide skill selection.** (a) A set of skill trajectories. (b) Naively selecting the skill whose terminal state is most distant from others. (c) Selecting the skill through the random walk process.

state is near potentially unexplored states, we utilize a simple random walk process (Fig. 2c). To elaborate, given a set of $P$ skills, (i) rollout skill trajectory ($T$ timesteps) and perform $R$ random walks from the terminal state of each skill, repeated $M$ times (i.e., a total of $P(T + R)M$ steps, collecting $PRM$ random walk arrival states as in Fig. 2c). Then (ii) we pinpoint the state in the lowest density region among collected random walk arrival states and select the skill which that state originated from as the guide skill $z^*$ (red skill in Fig. 2c). For (ii), one could use any algorithm to measure the density of the random walk state distribution. For our experiments, we utilize a simple k-nearest neighbor algorithm,

$$
z^* := \underset{p \in \{1,\ldots,P\}}{\mathrm{argmax}} \; \underset{r \in \{1,\ldots,RM\}}{\max} \frac{1}{k} \sum_{s_{pr}^j \in N_k(s_{pr})} ||s_{pr} - s_{pr}^j||_2
$$

$$
\text{where } s_{pr} = r\text{-th random walk arrival state of skill } p
$$

$$
N_k(\cdot) = \text{k-nearest neighbors.}
$$

(4)

In practice, in an environment with a long horizon (i.e., high $T$), such as DMC, our random walk process may cause sample inefficiencies. Therefore, we also present an alternative, efficient random walk process, which is thoroughly detailed in Appendix B. Ablation experiment on two guide skill selection methods (Fig. 2a,b) is provided in Section 4.4.

**Apprentice skills.** We select *apprentice skills* as the skills with low discriminability (i.e., skill $z^i$ with low $q_\phi(z^i|s)$; which fails to converge) and move them towards the guide skill. If most of the skills are already well discriminable (i.e., high MI rewards), we simply add new skills and make them apprentice skills. This will leave converged skills intact and send new skills to unexplored states. Since the total number of skills is gradually increasing as the new skills are added, this would bring the side benefit of not relying on a pre-defined number of total skills as a hyperparameter.

In Appendix E, we empirically show that adding new skills during training is generally difficult to apply to previous algorithms because the new skills will face pessimistic exploration problems. The new skills simply converge by overlapping with existing skills (e.g., left lower room in Fig 3), which exacerbates the situation (i.e., reducing discriminator accuracy without increasing state coverage).

## 3.2 Providing direct guidance via auxiliary reward

After selecting the guide skill $z^*$ and apprentice skills, we now formalize the exploration objective, considering two different aspects: (i) the objective of the guidance, and (ii) the degree of the guidance. It is crucial to account for these desiderata since strong guidance will lead apprentice skills to simply imitate guide skill $z^*$, whereas weak guidance will not be enough for skills to overcome pessimistic exploration problem.

In response, we propose an exploration objective that enables our agent to learn with guidance. We integrate these considerations into a single soft constraint as

$$\underset{\theta}{\text{maximize}} \ E_{z^i \sim p(z), s \sim \pi_\theta(z^i)} \left[ r_{\text{skill}} + r_{\text{guide}} \right]$$

$$\text{where } r_{\text{skill}} = \log q_\phi(z^i|s) - \log p(z^i), \tag{5}$$

$$r_{\text{guide}} = -\alpha \, \mathbb{I}\big(q_\phi(z^i|s) \leq \epsilon\big) \, (1 - q_\phi(z^i|s)) \, D_{\text{KL}}(\pi_\theta(a|s, z^i) || \pi_\theta(a|s, z^*)).$$

As we describe in Section 3.1, we select skills with low discriminator accuracy (i.e., $\mathbb{I}(q_\phi(z^i|s) \leq \epsilon)$) as apprentice skills. For (i), we minimize the KL-divergence between the apprentice skill and the guide skill policy (i.e., $D_{\text{KL}}(\pi_\theta(a|s, z^i) || \pi_\theta(a|s, z^*))$). For (ii), the extent to which the apprentice skills are encouraged to follow $z^*$, can be represented as the weight of the KL-divergence. We set the weight as $1 - q_\phi(z^i|s)$ to make the skills with low discriminator accuracy be guided more.

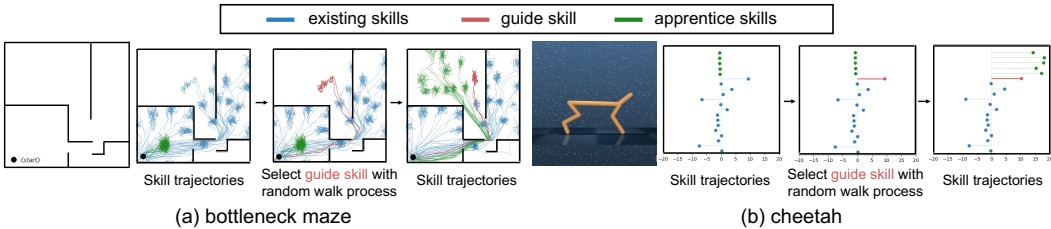

Figure 3: **Overall procedure of DISCO-DANCE** in (a) navigation and (b) continuous control tasks.

Fig. 3 shows that the guidance strategy significantly enhances the learning of skills in both navigation and continuous control environments. In the navigation task (Fig. 3a), the skill located in the upper left corner is selected as a guide through the random walk and utilized to navigate the apprentice skills, which were previously stuck in the lower left corner, into unexplored states. Similarly, in a continuous control environment (Fig. 3b), the skill that has been learned to *run* is selected as the guide, leading the apprentice skills that were barely moving. This guidance allows the apprentice skills to quickly learn to *run*. This salient observation suggests that the concept of guidance can be utilized, even for non-navigation tasks. In Section 4.4, we show that all of these components of $r_{\text{guide}}$ are necessary through additional experiments. We provide pseudocode for DISCO-DANCE in Algorithm 1.

---

**Algorithm 1:** Skill Discovery through Guidance

---

1   **Initialize** skills $z^1, ..., z^n$, RL policy $\pi_\theta(a|s, z)$, skill discriminator $q_\phi(z|s)$,
2   **Initialize** guide skill $z^* = $ None , $r_{\text{guide}} = 0$
3   **Hyperparameters** guide coef $\alpha$, apprentice cutoff $\epsilon$
4   **for** $k = 1, 2, ...$ **do**
5      Sample batch $(s_t, a_t, s_{t+1}, z^i)$ from replay buffer $D$
6      **if** *most skills are discriminable enough* **then**
7         guide skill $z^* \leftarrow$ *find_guide_skill*$(\pi_\theta, z^1, ..., z^n)$
8      $r_{\text{skill}} = \log q_\phi(z^i|s_{t+1}) - \log p(z^i)$
9      **if** $z^*$ *is not None* **then**
10        $r_{\text{guide}} = -\mathbb{I}\big(q_\phi(z^i|s) < \epsilon\big)(1 - q_\phi(z^i|s)) \, D_{\text{KL}}(\pi_\theta(a_t|s_t, z^i) || \pi_\theta(a_t|s_t, z^*))$
11      $r = r_{\text{skill}} + \alpha \cdot r_{\text{guide}}$
12      Update $\pi_\theta$ to maximize sum of $r$
13      Update $q_\phi$ to maximize $\log q_\phi(z|s_t)$

---

## 4   Experiments

### 4.1   Experimental Setup

**Evaluation.** We evaluate DISCO-DANCE across three distinct types of environments commonly used in USD: 2D Maze, Ant Maze, and Deepmind Control Suite (DMC). We utilize state space

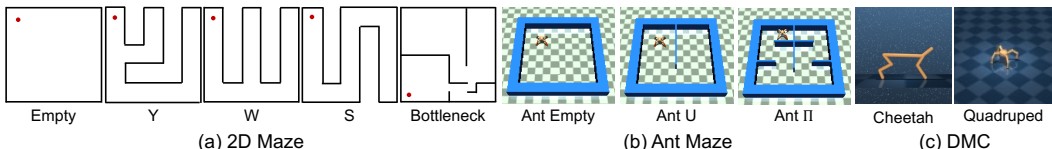

Figure 4: **Three environments to evaluate skill discovery algorithms.** (a) Continuous 2D mazes with various layouts, (b) high dimensional ant navigation tasks, and (c) continuous control environments with diverse downstream tasks.

coverage and downstream task performance as our primary evaluation metrics. To measure *state coverage*, we discretize the x and y axes of the environment into 10 intervals (i.e., total $10 \times 10$ buckets) and count the number of buckets reached by learned skills (Fig. 4.a,b). For *downstream task performance*, we finetune the agent pretrained with baselines and DISCO-DANCE (Fig. 4. (b,c)). Further details regarding the environments and evaluation metrics can be found in Appendix D.

For Ant mazes, we provide the (x,y) coordinates as the input for the discriminator for all algorithms following previous work [11, 47], since the main objective of navigation environments is to learn a set of skills that are capable of successfully navigating throughout the given environment. For DMC, we utilize all observable dimensions (e.g., joint angles, velocities) to the input of the discriminator to learn more general behaviors (e.g., running, jumping, and flipping) that can be used as useful primitives for unknown continuous control tasks.

**Baselines.** We compare DISCO-DANCE with various skill-based algorithms, focusing on addressing the pessimistic exploration of USD agents. We compare DISCO-DANCE with DIAYN [11], which trains an agent using only $r_{skill}$ without any exploration rewards. We also evaluate the performance of SMM [24], APS [25], and DISDAIN [52], which incorporate auxiliary exploration rewards. We note that all baselines and DISCO-DANCE utilize discrete skills, except APS, which utilizes continuous skills. For downstream tasks, we also compare USD baselines with SCRATCH, which represents *learning from scratch* (i.e., no pretraining). Additional details are available in Appendix D.

In addition, we include UPSIDE [19] as a baseline. UPSIDE learns a tree-structured policy composed of multiple skill nodes. Although UPSIDE does not utilize auxiliary reward, DISCO-DANCE and UPSIDE both aim to enhance exploration by leveraging previously discovered skills. However, UPSIDE's *sequential execution* of skills from ancestors to children nodes in a top-down manner leads to significant inefficiency during finetuning, leading to lower performance on downstream tasks. Furthermore, UPSIDE selects the skill with *highest discriminator accuracy* (i.e., corresponding to Fig. 2b) for expanding tree policies, resulting in reduced state coverage at the pretraining stage. A detailed comparison between DISCO-DANCE and UPSIDE can be found in Appendix F.

### 4.2 Navigation Environments

#### 4.2.1 2D Mazes

First, we evaluate DISCO-DANCE in 2D mazes (Fig. 4a), which has been commonly used for testing the exploration ability of skill learning agents [7, 19]. The layout becomes more challenging, from an empty maze to a bottleneck maze. The agent determines where to move given the current x and y coordinates (2-dimensional state and action space).

Table 1: **State space coverages of DISCO-DANCE and baselines on two navigation benchmarks.** The results are averaged over 10 random seeds accompanied by a standard deviation. Scores in bold indicate the best-performing model and underlined scores indicate the second-best.

| Models | (a) 2D mazes | | | | | (b) Ant mazes | | |
|---|---|---|---|---|---|---|---|---|
| | Empty | Y | W | S | Bottleneck | Ant Empty-maze | Ant U-maze | Ant Π-maze |
| DIAYN | **100.00**±**0.00** | 69.80 ±5.39 | 71.50 ±5.10 | 52.80±5.13 | 52.40±3.77 | 72.70±10.95 | 46.80±8.41 | 22.50±3.34 |
| SMM | **100.00**±**0.00** | 89.20 ±5.80 | 73.40 ±6.15 | 55.00±5.59 | 54.80±5.11 | **99.50**±**0.84** | 58.90±8.23 | 25.40±4.97 |
| APS | 97.90±3.75 | 90.00 ±5.51 | 81.50 ±11.31 | 80.20±5.88 | 61.90± 10.55 | 83.10±24.80 | 59.60±5.85 | 26.3±7.68 |
| DISDAIN | **100.00**±**0.00** | 87.20 ±6.81 | 85.00 ±8.69 | 61.30±7.04 | 61.70±4.85 | 70.10±4.97 | 45.80±4.98 | 22.90±2.88 |
| UPSIDE | 99.00±9.25 | 88.20 ±19.39 | 90.20 ±6.15 | 69.90±7.63 | 78.90±13.69 | 74.50±6.22 | 63.60±11.58 | 29.90±2.42 |
| DISCO-DANCE | **100.00**±**0.00** | **99.10** ±**1.66** | **98.30** ±**2.62** | **88.50**±**6.45** | **86.30**±**17.01** | 98.90±1.85 | **68.30**±**4.47** | **39.00**±**4.85** |

Table 1(a) summarizes the performance of skill learning algorithms on 2D mazes. Our empirical analysis indicates that baseline methods with auxiliary reward (i.e., SMM, APS, and DISDAIN) exhibit superior performance compared to DIAYN. However, we find out that previous studies provide less benefit on performance as the layout becomes more complex, as we mentioned in section 2.2. In contrast, the performance of DISCO-DANCE remains relatively stable regardless of layout complexity. Additionally, DISCO-DANCE outperforms UPSIDE in all 2D mazes, which we attribute to the inefficiency of adding new skills to leaf nodes with high discriminator accuracy (Fig. 2b).

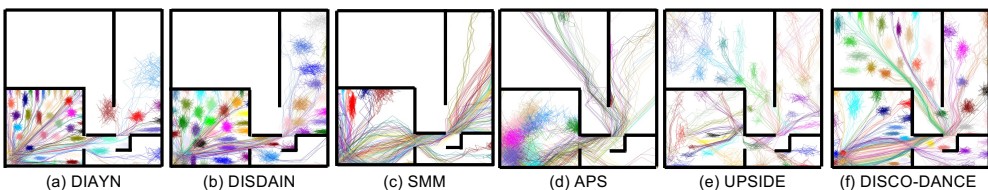

|  (a) DIAYN | (b) DISDAIN | (c) SMM | (d) APS | (e) UPSIDE | (f) DISCO-DANCE |

Figure 5: **Visualization of the skills in bottleneck maze.** Multiple rollouts by each algorithm.

Fig. 5 illustrates multiple rollouts of various skills learned in the 2D bottleneck maze. For the 2D bottleneck maze, the upper left room is the most difficult region to reach since the agents need to pass multiple narrow pathways. Fig. 5 shows that only DISCO-DANCE and UPSIDE effectively explore the upper left region, with DISCO-DANCE displaying a relatively denser coverage on the upper left state space. More qualitative results are available in Appendix I.

### 4.2.2 Ant mazes

To evaluate the effectiveness of DISCO-DANCE in the environment with high dimensional state and action space, we train DISCO-DANCE in three Ant mazes where the state space consists of joint angles, velocities, and the center of mass, and the action space consists of torques of each joint [8, 30].

Table 1(b) reports the state coverage of DISCO-DANCE and baselines. DISCO-DANCE shows superior performance to the baselines where it achieves the best state-coverage performance in high-dimensional environments that contain obstacles (i.e., Ant U-maze and $\Pi$-maze) and gets competitive results against SMM with a marginal performance gap in the environment without any obstacle (i.e., Ant Empty-maze). These results demonstrate the effectiveness of our proposed direct guidance in navigating an environment with obstacles. Additional qualitative results of each algorithm can be found in the Appendix I.

To further assess whether the learned skills could be a good starting point for downstream tasks, we conduct goal-reaching navigation experiments where the goal state is set to the region farthest from the initial state for the more challenging navigation environment, Ant $\Pi$-maze. We measure the number of seeds that successfully reach the established goal state (out of a total 20 seeds). We select the skill with the maximum return (i.e., a skill whose state space is closest to the goal state) from the set of pretrained skills and finetune the policy $\pi_\theta(a|s,$ selected skill $z)$. Additional experiments in Ant U-maze and details are available in Appendix D, E.

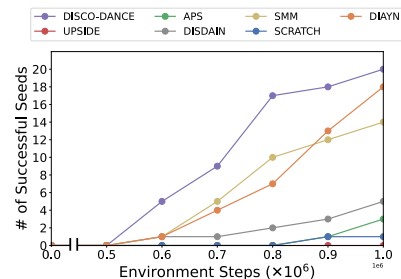

Figure 6: **Finetuning results on the Ant $\Pi$-maze**. We plot the number of successful seeds out of total 20 seeds.

Fig. 6 shows that DISCO-DANCE outperforms prior baseline methods in terms of sample efficiency. Specifically, DISCO-DANCE is the only algorithm that successfully reaches the goal state for all seeds (20 seeds) in Ant $\Pi$-maze. In addition, UPSIDE fails to reach the goal for all seeds, which indicates that the tree-structured policies learned during the pretraining stage can impede the agent from learning optimal policies for fine-tuning.

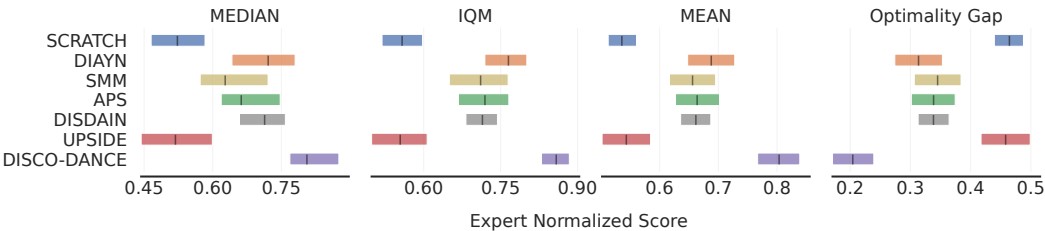

Figure 7: **Performance comparisons on Deepmind Control Suite.** Aggregate statistics and performance profiles with 95% bootstrap confidence interval are provided [2], which are calculated across 120 seeds (15 seeds × 8 tasks).

## 4.3 Deepmind Control Suite

While the experiments in 2D and Ant mazes demonstrate that the skills DISCO-DANCE learns can be utilized as useful primitives for navigation tasks, to demonstrate the versatility and effectiveness of DISCO-DANCE in acquiring *general skills* (e.g., run, jump, flip), we conduct additional experiments on the DMC benchmark [53]. Specifically, we consider two different environments in DMC, Cheetah, and Quadruped, and evaluate DISCO-DANCE across four tasks (e.g., run, flip, and jump) within each environment (total of 8 tasks). The DMC benchmark evaluates the diversity of the learned skills, meaning that agents must learn a suitable set of skills for all downstream tasks in order to achieve consistently high scores across all tasks. We first perform pretraining for 2M training steps and select the skill with the highest return for each task (same as the Ant maze experiment). Then we finetune USD algorithms for 100k steps. More experimental details are available at Appendix D.

Fig. 7 shows the performance comparisons on DMC across 8 tasks with 15 seeds. First, all USD methods perform better than training from scratch (i.e., SCRATCH), indicating that pretraining with USD is also beneficial for continuous control tasks. Furthermore, as we described in Fig. 3b, we find that our proposed *guidance* method boosts the skill learning process for the continuous control environment, resulting in DISCO-DANCE outperforming all other baselines. As shown in Ant mazes finetuning experiments, UPSIDE significantly underperforms compared to other auxiliary reward-based USD baselines. Since UPSIDE necessitates the sequential execution of all its ancestor skills, its finetuning process becomes less efficient than other methods.

In addition, baselines with auxiliary rewards, which enforce exploration, are ineffective in DMC. We attribute these results to their training inefficiency, that the current experimental protocol of pretraining for 2M frames is insufficient for them to converge in a meaningful manner. For example, DISDAIN requires billions of environment steps in the grid world environment in the original paper. In contrast, DISCO-DANCE successfully converges within 2M frames and outperforms all other baselines. We provide the full performance table and visualizations of skills in Appendix E, I.

## 4.4 Ablation Studies

### 4.4.1 Model Components

We further conduct an ablation analysis to show that each of our model components is critical for achieving strong performance. Specifically, we experiment with two variants of DISCO-DANCE, each of which is trained (i) without guide coefficient (i.e., guide coefficient as 1) and (ii) without the random walk based guide selection process in Table 2.

Table 2: **Ablation study on model components.**

| Models | Bottleneck maze |
|---|---|
| DISCO-DANCE | **86.30±17.01** |
| DISCO-DANCE w/o guide coef | 76.75±13.05 |
| DISCO-DANCE w/o random walk | 70.30±10.72 |
| DIAYN (no exploration reward) | 52.40±3.77 |

First, without guide coefficient $\mathbb{I}\big(q_\phi(z^i|s) \leq \epsilon\big)\,(1 - q_\phi(z^i|s))$, DISCO-DANCE sets all other skills as apprentice skills, causing all other skills to move towards the guide skill. As shown in Table 2, this method of guiding too many skills degrades performance, and it is important to only select unconverged skills as apprentice skills for effective learning.

Second, Table 2 also shows that random-walk based guide skill selection (i.e., selecting the skill with the lowest density region among PRM terminal states in Eq. 4) is superior to naively selecting the most distant skill without random walk process (i.e., selecting the skill with the lowest density region among P terminal states in Eq. 4). We also provide qualitative results (Fig. 8) where selecting the most distant skill as the guide skill fails to select the skill closest to the unexplored state region since the blue skill is the most distant from the terminal states of the other skills. On the other hand, random walk guide selection successfully finds the appropriate guide skill.

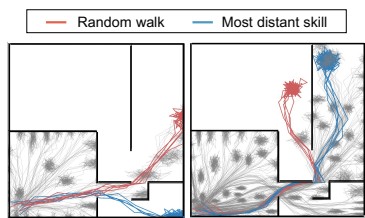

Figure 8: **Qualitative results for different guide skill selection processes.**

#### 4.4.2 Compatibility with other skill discovery method

In this paper, our primary goal is to compare DISCO-DANCE with algorithms that focus on constructing an effective auxiliary exploration reward, denoted as $r_{\text{exploration}}$. However, several research studies exist that seek to refine $r_{\text{skill}}$ to enhance exploration [32, 20, 27] (detailed explanation in Section 5). To evaluate how effectively DISCO-DANCE can enhance exploration when aligned with different $r_{\text{skill}}$, we carried out a comparative analysis with LSD [32]. The recently introduced skill discovery algorithm, LSD, has demonstrated commendable outcomes, especially in Ant Mazes.

Table 3: **Ablation study on $r_{\text{skill}}$.**

| $r_{\text{skill}}$ | $r_{\text{exploration}}$ | Ant $\Pi$-maze |
|---|---|---|
| DIAYN | - | 22.50 $\pm$3.34 |
|  | $r_{\text{guide}}$ | 39.00 $\pm$4.85 |
| LSD | - | 38.80 $\pm$3.34 |
|  | $r_{\text{guide}}$ | 45.80 $\pm$3.34 |

Table 3 illustrates the performance comparison with LSD. Our findings reveal that LSD, even in the absence of $r_{\text{exploration}}$, exhibits superior performance compared to DIAYN in Ant-$\Pi$ maze. Such an observation possibly highlights LSD's proficiency in mitigating the inherent pessimism linked with the previous mutual information objective. An observation worth highlighting is the substantial enhancement in performance for both algorithms when integrated with our $r_{\text{guide}}$. We propose that $r_{\text{skill}}$ and $r_{\text{guide}}$ have distinct roles, and when combined, they can collectively boost their overall performance.

## 5 Related Work

**Pretraining methods in RL** Pretraining methods for RL primarily fall into two categories [55]. The first is online pretraining, where agent interact with the environment in the absence of any reward signals. This method generally operates under the presumption that agents can engage with the environment at a minimal cost. The primary objective of online pretraining is to acquire essential prior knowledge through unsupervised interactions with the environment. Recently, there has been a lot of research in this area, often referred to as "Unsupervised RL". Such methodologies typically employ intrinsic rewards, to learn representations or skills which will be useful for downstream tasks. The acquired pretrained weights, such as encoder, actor, and critic, are subsequently reused for downstream tasks.

Depending on how the intrinsic reward is modeled, online pretraining can be further categorized. Curiosity-driven approaches focus on probing states of interest, often characterized by high predictive errors, in order to obtain more environmental knowledge and reduce prediction errors across states [6, 35, 43, 34]. Data coverage maximization methods aim to maximizes the diversity of the data the policy has collected, for instance, by amplifying the entropy of state visitation distributions [5, 18, 24, 26, 57, 44]. Meanwhile, Unsupervised Skill Discovery involves learning policies, denoted as $\pi(a|s,z)$, guided by a distinct skill vector $z$. This vector can then be adjusted or combined to address subsequent tasks [13, 11, 17, 47, 7, 25, 19, 32, 1, 60, 39, 33].

The second category is offline pretraining. This method leverages an unlabeled offline dataset accumulated through various policies, to learn the representation. The learning objective in offline pretraining involves pixel or latent space reconstruction [58, 59, 61, 54], future state prediction [41, 42, 45, 23], learning useful skills from data [37, 3, 50, 36, 16, 56] and contrastive learning focusing on either instance [21, 12] or temporal discrimination [31, 51].

In this paper, our primary emphasis is on Unsupervised Skill Discovery within the domain of online pretraining. For a comprehensive overview of pretraining methods for RL, we direct the reader to [22, 55].

**Unsupervised skill discovery.** There have been numerous attempts to learn a set of useful skills for RL agents in a fully unsupervised manner. These approaches can be broadly classified into two categories: (i) one studies how to learn the skill representations in an unsupervised manner (i.e., how to make $r_{\text{skill}}$ in Eq. 5) and (ii) the other focuses on devising an effective auxiliary reward which encourages the exploration of skills to address the inherent pessimistic exploration problem (i.e., how to make $r_{\text{exploration}}$ in Eq. 5).

Research that falls under the first category focuses on how to train skills that will be useful for unseen tasks. DIAYN [11], VISR [17], EDL [7] and DISk [46] maximizes the mutual information (MI) between skill and state reached using that skills. DADS [47] utilizes a model-based RL approach to maximize conditional MI between skill and state given the previous state. LSD [32], which is non-MI-based algorithm, provides a 1-Lipschitz constraint on the representation of skills to learn dynamic skills. CIC [20] replace $r_{\text{skill}}$ with a contrastive loss to learn high-dimensional latent skill representations, and additionally utilizes $r_{\text{exploration}}^{\text{APS}}$ (Eq. 3) for exploration. Choreographer [27] learns a world model to produce imaginary trajectories (which are used for learning skill representations) and additionally utilizes $r_{\text{exploration}}^{\text{APS}}$ for exploration.

DISCO-DANCE belongs to the second category that focuses on devising exploration strategy for USD. This category includes DISDAIN [52], UPSIDE [19], APS [25] and SMM [24], which are included as our baseline methods. Detailed explanation and limitations of these methods are described in Section 2.2, 4.1, Fig. 1, and Appendix F.

**Guidance based exploration in RL.** Go-Explore [9, 10] stores visited states in a buffer and starts re-exploration by sampling state trajectory, which is the most novel (i.e., the least visited) in the buffer. DISCO-DANCE and Go-Explore share a similar motivation that the agent guides itself: DISCO-DANCE learns from other skills and Go-Explore learns from previous trajectories. Another line of guided exploration is to utilize a KL-regularization between the policy and the demonstration or sub-policy (i.e., guide) [37, 38, 8, 29, 48]. SPIRL [37], an offline skill discovery method, minimizes the KL divergence between the skill policy and the learned prior from offline data. RIS [8] has been proposed for efficient exploration in goal-conditioned RL, by minimizing KL divergence between its policy and generated subgoals.

# 6   Conclusion

We introduce DISCO-DANCE, a novel, efficient exploration strategy for USD. It directly guides the skills towards the unexplored states by forcing them to follow the *guide skill*. We provide quantitative and qualitative experimental results that demonstrate that DISCO-DANCE outperforms existing methods in two navigation and a continuous control benchmark.

In this paper, in order to evaluate solely the effectiveness of $r_{\text{exploration}}$ methods, we fix the backbone $r_{\text{skill}}$ as DIAYN. Combining DISCO-DANCE with other recent $r_{\text{skill}}$ methods (e.g., CIC) is left as an interesting future work. Also, we think that there is still enough room for improvement in finetuning strategies [22]. Since USD learns many different task-agnostic behaviors, a new fine-tuning strategy that can take advantage of these points would make the downstream task performances more powerful. A discussion of limitations is available in Appendix G.

## Acknowledgments and Disclosure of Funding

This work was supported by the Institute of Information & communications Technology Planning & Evaluation (IITP) grant funded by the Korea government(MSIT) (No.2019-0-00075, Artificial Intelligence Graduate School Program(KAIST)), the National Research Foundation of Korea (NRF) grant funded by the Korea government (MSIT) (No. NRF-2022R1A2B5B02001913), and the National Supercomputing Center with supercomputing resources including technical support (KSC-2023-CRE-0074).

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

# Appendix

## A  Implementation Details of DISCO-DANCE

For reproducibility, we (i) provide pseudo code in Algorithm 1 and (ii) open-source code at `https://mynsng.github.io/discodance/`. Below, We describe further details regarding the implementation of DISCO-DANCE.

- (line 1 in Algorithm 1) When extending new skills, we initialize their policy weights with the guide skill policy weights rather than random weights as in SPIRL [37]. This accelerates exploration by encouraging the newly added skills to directly reach the unexplored regions. In Appendix E.3, we conduct ablation studies of this component.

- (line 4 in Algorithm 1) As the number of skills is gradually increasing, the discriminator struggles to learn due to the sparse update for each skill by the decreased probability of sampling each skill. We mitigate this issue by utilizing a skill sampling scheme where each skill sampling weight is proportionate to its discriminator error (e.g., skills with high discriminator error are more likely to be sampled). This approach is analogous to the methodology adopted in Prioritized Experience Replay [40].

- (line 5-6 in Algorithm 1) For adding new skills, we need to define hyperparameters: (1) when to extend skills and (2) how many to increase. We gradually extend skills as the discriminator converges. In detail, count the number of skills that have low discriminator accuracy (i.e., less than $\epsilon$), and if the number is less than the threshold (i.e., $\rho$), extend $\rho$ skills.

- (line 8-9 in Algorithm 1) We utilize a KL divergence between the apprentice skills and the guide skill policy as $r_{\text{guide}}$. However, as the guide skill policy changes during training, it can lead to instability in the training of the apprentice skills. To address this issue, we use a separate target policy network for the guide skill. More precisely, when we select the guide skill, we copy and fix the policy network (denoted as target policy network $\hat{\theta}$) and calculate the KL divergence reward utilizing this target policy network (i.e., calculating $r_{\text{guide}}$ as $-\mathbb{I}\big(q_\phi(z^i|s) < \epsilon\big)(1 - q_\phi(z^i|s))D_{\text{KL}}(\pi_\theta(a_t|s_t, z^i)||\pi_{\hat{\theta}}(a_t|s_t, z^*)))$.

- (line 13-14 in Algorithm 1) Entropy term in SAC can be seen as KL divergence between the policy and uniform distribution with a constant [37] ($\mathcal{H}(\pi(\cdot|s_t, z_t)) \propto -D_{KL}(\pi(\cdot|s_t, z_t)||U(\cdot))$. Instead of replacing the entropy term with guide term in DISCO-DANCE, we found that using both term simultaneously helped stabilize learning by enjoying the advantage of maximum entropy RL (i.e., RL agent maximizes $E_\pi[\sum_{t=1}^T \gamma^t(r_{\text{skill}} + \alpha\, r_{\text{guide}} + \beta\, H(\pi(\cdot|s_t)))])$.

## B   Efficient Random Walk Process

Our random walk process in Section 3 is as follows:

1. After performing rollout for all $P$ learned skills till the terminal state, we perform $R$ random walks and collect $P * R$ number of random arrival states ($R$ is a hyper-parameter and we set $R$ as 0.2 * time horizon $T$ for all environments). Repeat this process a total of $M$ times.

2. Pinpoint the state in the lowest density region among the $P * R * M$ number of states. We measure the density of the states using the k nearest neighbors.

3. Select the skill which that state originated from.

The number of environment steps in the random walk process is $P * (T + 0.2T) * M$. Therefore, the total number of environment steps performed during the random walk process in pretraining stage is,

$$(N_{\text{initial}} + (N_{\text{initial}} + \delta) + (N_{\text{initial}} + 2 * \delta) + .. + (N_{\text{final}} - \delta)) * (T + 0.2T) * M$$

where $N_{\text{initial}}$ is the number of initial skills, $\delta$ is the number of skills to extend when most of the existing skills are converged (e.g., number of skills are $10 \rightarrow 15 \rightarrow 20 \rightarrow \ldots \rightarrow 50 \rightarrow 55$ when $N_{\text{initial}}$=10, $\delta$=5, $N_{\text{final}}$=55). Note that $N_{\text{final}}$ is the number of skills when the pretraining is ended (not predefined as a hyperparameter).

However, in long-horizon environment such as DMC (1000 timesteps), our random walk process can cause non-negligible sample inefficiencies. Thus, we further present an efficient random walk process that approximates an original random walk process with a simple approach. The detailed process is as follows:

1. During pretraining, if the skill satisfies $q_\phi(z^i|s_T) < \epsilon$ in the terminal state $s_T$ (i.e., which is not an apprentice skill), additionally perform $R$ random walk steps and store the random arrival states at the *temporary buffer (queue)*.

2. When selecting guide skill, instead of rolling out all skills, select the state in the lowest density region among the states at the *temporary buffer*.

3. Select the skill which that state originated from.

Since there are no additional environment steps when selecting the guide skill, the efficient random walk process can significantly reduce the number of environment steps compared to the original version (added environment steps are $0.2T$ per trajectory (only if its skill satisfies the accuracy threshold), whereas original random walk process requires $P \times (T + 0.2T) \times M$). Our experimental results on DMC demonstrate that the efficient random walk process outperforms other baselines.

# C  Random Walk Process in High-dimensional State Space

To evaluate the scalability of our random walk process within high-dimensional state spaces, we've identified two essential questions to consider:

1. When the agent starts to explore from an arbitrary terminal state, can the agent visit a diverse range of states through the random walk process?

2. Given the diverse range of visited states, is it possible to identify the terminal state (guide skill) within the least explored region?

In response, we conducted a synthetic experiment on Montezuma's Revenge, which is a high-dimensional pixel-based environment characterized by an 84×84×3 input and is well-known for its exploration challenges.

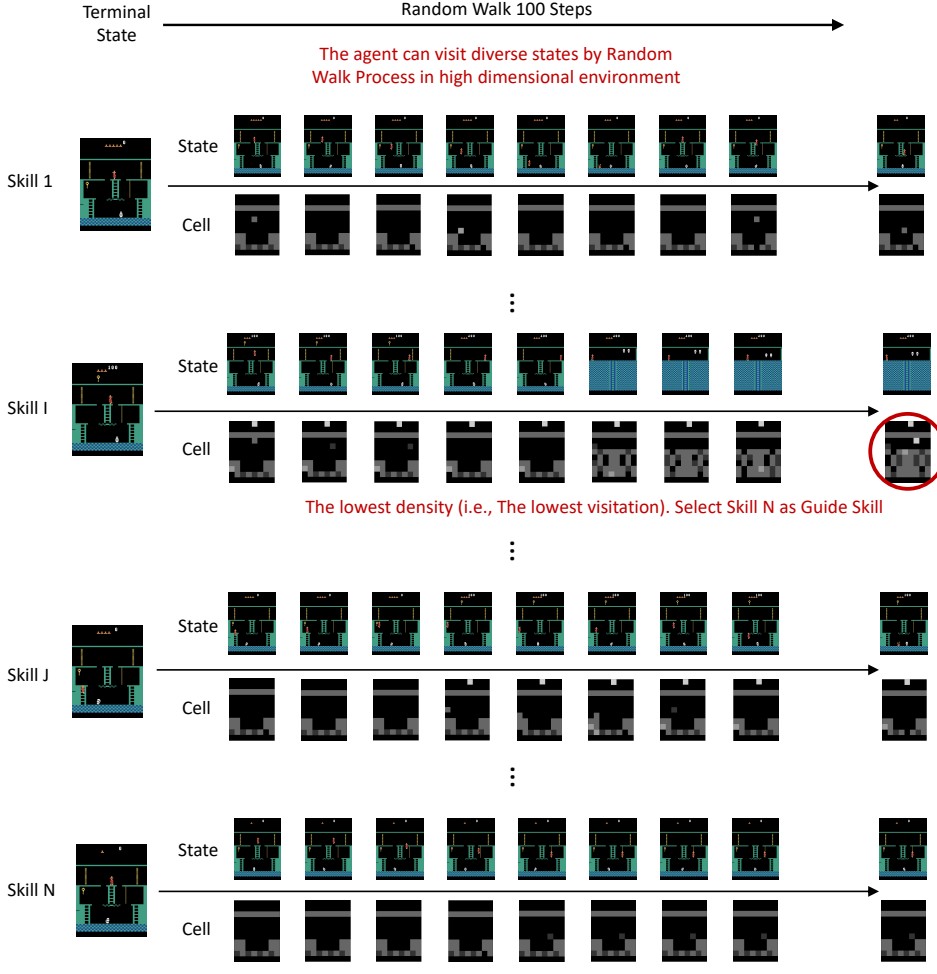

Figure 9: **Random walk process in high dimensional environment**

Figure 9 provides a visual illustration of our random walk approach within Montezuma's Revenge. In this experiment, we:

1. Randomly reset the agent in the initial room of Montezuma's Revenge.

2. Execute a random walk for 100 steps from the randomly initialized state.

3. Iterate the step 1,2 for $N$ cycles.

This experimental design mirrors the algorithmic design of DISCO-DANCE where each reinitialized point indicates the terminal state of a different skill. Correspondingly, each skill undergoes a random walk for 100 steps, aligning with the parameters used for our paper (i.e., P=N, R=100, M=1 in Equation 4).

Regarding the first question, as illustrated by Figure 9, even in such a high-dimensional state space, the agent was able to visit a variety of states within just 100 random steps. For instance, for Skill 'I', the agent was able to move to another room. And with for Skill 'J', the agent successfully picked up the key. This experimental result supports the versatility of the random walk process, even when the environment is high-dimensional.

Regarding the second question, we employed the cell-centric technique from Go-Explore [1] for density estimation: (i) segmenting the aggregated states into discrete cells, (ii) count the number of each cell's visitation, and (iii) select the least visited cell. As emphasized in Figure 9, the cell marked in red is selected as guide skill, indicating that skill 'I' is the prime candidate for guide skill in DISCO-DANCE.

This combined approach of exploration through random walk then identifying unique states (which is used in DISCO-DANCE), parallels the approach adopted by Go-Explore. This technique has consistently demonstrated its efficacy across varied domains, including Atari games and robotic settings. In summary, we believe our synthetic experiment affirms the scalability of our random walk process in DISCO-DANCE, even within a high-dimensional pixel-based environment.

# D    Experimental Details

## D.1    Environments

**2D maze** The width and height of easy and medium level mazes are both 5, and 10 for hard level maze [7]. There is no explicit terminal state, and the episode ends only when the maximum timestep (30 for easy and medium levels and 50 for hard level) is reached. The state space and action space are both 2-dimensional (state: x, y coordinates, action space: distance to move in x, y coordinates).

**Ant maze** The width and height of U-maze are (7,7) and Empty, Π-shaped maze are (8,8) respectively [8, 30]. Similar to 2d-maze, episode ends only when the maximum timestep (400 for every mazes) is reached. The state space is 31-dimensional, and the action space is 8-dimensional.

**Deepmind Control Suite** We conduct our environment in a total of eight downstream tasks provided by URLB [22]. In detail, we utilize four tasks each in two continuous control domains: Run, Run backward, Flip, and Flip backward for Cheetah and Jump, Run, Stand, and Walk for Quadruped. For cheetah, the state space is 17-dimensional, and the action space is 6-dimensional. For Quadruped, the state space is 78-dimensional, and the action space is 12-dimensional.

## D.2    Evaluation

**State Coverage (2D, Ant mazes)** We discretize the x and y axes of the environment into 10 intervals (i.e., total $10 \times 10$ buckets) and count the number of buckets reached by learned skills.

**Downstream Task Performance** For Ant mazes, we conduct goal-reaching downstream tasks. We design a distance between the current state and the goal state as a penalizing reward (i.e., $-1 \times \text{distance} = \text{reward}$). In detail, directly computing distance with x and y coordinates might be inaccurate because there can be obstacles (i.e., walls) between two states. Therefore, we design a reward function that returns the actual distance that goes around when there is an obstacle between two states. Then, we normalize the reward to set the lowest value as $-1$. For Ant mazes and DMC, we first perform pretraining for 5M(Ant mazes) / 2M(DMC) training steps and select the skill with the highest return for each task. Then we finetune for 1M(Ant mazes) / 100k(DMC) steps.

For DIAYN, SMM, DISDAIN, and DISCO-DANCE (discrete skills), we finetune the skill with maximum downstream task reward. For APS (continuous skill), we first randomly select an arbitrary skill z and rollout episodes, then solve the linear regression problem to select the task-specific skill z

(following the same protocol in the APS paper). After the skill is selected for finetuning, we initialize the agent with the pre-trained network parameters. Utilizing pretrained parameters for the entire parameters of the critic can impede the rapid adaptation process, as the pretrained network is already optimized for the intrinsic reward. To mitigate this issue, we initialize only the encoder of the critic network with pretrained parameters (i.e., excluding the last fully connected layer). In addition, we initialize the actor network with entire pretrained parameters (including the last fully connected layer).

Note that our evaluation on the downstream task is different from URLB in that the skill selection process is not included in 100K, but this is still a fair comparison as all algorithms use the same number of interactions (samples) to select skills. For UPSIDE, (i) first we select the leaf node skill with maximum downstream task rewards, (ii) then freeze all the pretrained ancestors of that selected leaf node skills and (iii) only finetune the leaf node skill, same as the original paper. A detailed explanation on the fine-tuning procedure of UPSIDE is described in Appendix F

### D.3 Hyperparameters

Detailed hyperparameters of our method DISCO-DANCE are listed in the table 4. Note that the additional steps used to perform the random walk process have been included in the overall count of environment steps for training.

Table 4: **A common set of hyperparameters used in DISCO-DANCE**.

| Hyperparameter | Value |
|---|---|
| Replay buffer size | $10^6$ |
| Optimizer | Adam |
| Learning rate | $3 \times 10^{-4}$ |
| RL algorithm | Soft Actor Critic |
| discount $\gamma$ | 0.99 |
| Select guide | K nearest neighbor |
| Guide coef ($\alpha$) | $10^{-4}$ |
| Extending threshold $\rho$ | 5 |
| Accuracy threshold $\epsilon$ | 0.5 |

Table 5: **Per environment sets of hyperparameters used in DISCO-DANCE.**

| Hyperparameter | 2D maze | Ant maze | DMC |
|---|---|---|---|
| Hidden dim | 128 | 1024 | 1024 |
| Batch size | 64 | 512 | 1024 |
| Skill trajectory length | easy, normal: 30 hard: 50 | 400 | 1000 |
| Initial number of skill | 30 | 30 | 10 |
| Entropy coef ($\beta$) | 0.2 | 0.6 | 0.2 |
| Number pre-training frames | easy, normal: $2 \times 10^6$ hard: $5 \times 10^6$ | $5 \times 10^6$ | $2 \times 10^6$ |

For DISDAIN, we search the number of ensembles from [2, 10, 20, 40] and disdain reward coefficient from [10, 50, 100]. For 2D maze, We train DISDAIN using 40 sizes of ensembles and set DISDAIN reward weight as 100. For Antmaze, we set 20 for the number of ensembles and 100 for the reward coefficient. For DMC, we utilize 2 for the number of ensembles and 10 for the reward coefficient.

For APS, we search the value of k (i.e., the number of neighbors in k nearest neighbor algorithm) from [5, 10, 12] and the dimension of skills from [5, 10]. For 2D maze, we utilize 10 for k and 5 for skill dimensions. For Antmaze, we utilize 5 for k and 5 for skill dimensions. For DMC, we utilize 12 for k and 10 for skill dimensions.

For SMM, we search the entropy coefficient from [0.0005, 0.005, 0.01, 0.1, 0.25, 0.5, 1]. We utilize the entropy coefficient as 0.1 for 2D maze and Antmaze, and 0.01 for DMC.

For UPSIDE, there are five kinds of hyperparameters: maximum timestep $T$, diffusing timestep $H$, discriminator threshold $\eta$, initial extend skill number $N_{start}$, and maximum extend skill number $N_{max}$. We set $T = H = 5$ for 2D maze, $T = H = 50$ for Antmaze and $T = H = 125$ for DMC. In addition, we utilize $\eta = 0.5$, $N_{start} = 2$, and $N_{max} = 8$ for all environments.

We use SAC as the backbone RL algorithm [15]. We train DIAYN, APS, and SMM using the code provided by URLB [22]. In addition, we re-implement DISDAIN and UPSIDE by strictly following the details of the paper.

We use a fixed number of skills for the baselines (i.e., DIAYN, DISDAIN, and SMM), which utilize discrete skill spaces. We set 50 for easy and normal level 2D mazes, 70 for the hard level 2D maze, and 100 for Antmaze and DMC.

### D.4 Resource Requirements

Full experimental training runs 12 hours for the 2D maze and 24 hours for Antmaze and DMC. We conducted all experiments using a single RTX 3090 GPU, and each experiment requires approximately up to 2GB of memory.

### D.5 License

- Environment
    - 2D mazes: We used the publicly available environment code in [7][2].
    - Ant mazes: We used the publicly available environment code in [8][3].
    - Deepmind Control suite: We used the publicly available environment code in [22, 53][4].
- Baseline algorithm
    - For DIAYN, APS and SMM, we used the code provided in [22].

## E  Additional Experimental Results

### E.1  Additional fine-tuning results on the Ant U-maze

To further evaluate whether the learned skills can be useful for downstream tasks, we conducted goal-reaching navigation experiment in Ant U-maze, wherein the goal state was designated as the region positioned furthest from the initial state. The quantification of successful goal attainment was measured by determining the number of seeds, out of a total of 20, in which the agents reached the goal state. We select the skill with the maximum return (i.e., skill whose state space is closest to the goal state) from the set of pretrained skills, and finetune the policy. Notably, DISCO-DANCE reached the goal in the Ant U maze faster than the other algorithms (as indicated by the earliest point where the y-axis reaches 20).

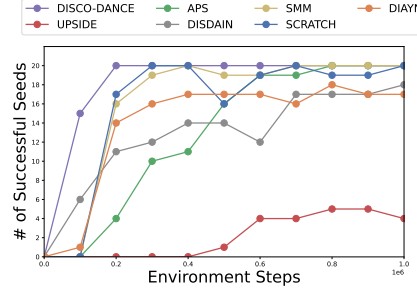

Figure 10: **Finetuning results on the Ant U-maze.** We plot the number of successful seeds out of total 20 seeds.

Fig. 10 shows that DISCO-DANCE consistently reaches the goal state (for all 20 seeds) with the fewest number of environmental steps. As shown in the Ant $\Pi$-maze finetuning experiment (Section 4), the performance of UPSIDE is significantly inferior to the performance of DISCO-DANCE, with only 5 seeds out of 20 seeds reaching the goal state. These results suggest that the tree-structured policies during the pretraining stage hinders the agent's ability to learn optimal policies during the fine-tuning phase.

---

[2] https://github.com/victorcampos7/edl
[3] https://github.com/elliotchanesane31/RIS
[4] https://github.com/rll-research/url_benchmark

## E.2 Full Results on Deepmind Control Suite Tasks

Table 6: Performance comparison of DISCO-DANCE and baselines on DMC. Bold scores indicate the best model performance and underlined scores indicate the second best. The results are averaged over 15 random seeds with a standard deviation.

| Models | Cheetah | | | | Quadruped | | | | Avg |
|---|---|---|---|---|---|---|---|---|---|
| | Run | Run Backward | Flip | Flip Backward | Jump | Run | Stand | Walk | |
| SCRATCH | 411.25±58.24 | 386.20±16.35 | **704.54±16.84** | 714.10±8.62 | 345.05±196.17 | 140.64±80.13 | 552.07±224.75 | 119.90±171.63 | 421.71 |
| DIAYN | 532.63±143.12 | **450.35±8.90** | 571.16±164.78 | 717.72±20.01 | 629.26±217.02 | 414.89±230.47 | 743.53±257.33 | 367.65±344.14 | 553.40 |
| SMM | 580.79±54.92 | 436.35±13.04 | 668.12±169.16 | **733.79±16.86** | 460.38±245.73 | 342.51±214.29 | 434.17±266.74 | 511.40±330.69 | 520.94 |
| APS | 574.00±56.47 | 437.66±16.96 | 640.51±80.81 | 684.86±48.82 | 572.32±247.61 | 337.20±217.22 | 640.51±279.73 | 366.45±305.55 | 531.69 |
| DISDAIN | 586.44±59.24 | 445.85±15.87 | 673.54±89.86 | 731.95±8.24 | 476.16±200.91 | 246.45±124.32 | 725.42±200.08 | 347.24±179.32 | 529.13 |
| UPSIDE | 309.81±121.52 | 259.17±94.74 | 607.96±117.82 | 622.84±78.48 | 495.39±269.78 | 248.32±198.72 | 635.24±312.12 | 359.62±307.36 | 442.29 |
| DISCO-DANCE | **602.74±36.64** | 448.50±12.93 | 703.80±60.44 | 705.23±56.96 | **693.23±249.01** | **477.41±239.69** | **941.94±125.25** | **664.66±346.16** | **654.69** |

Table 6 summarizes the full results for each task in DMC. We can observe that DISCO-DANCE gets first or second performance in seven out of eight tasks. The performance of the USD baseline was superior to the *scratch*, demonstrating that previously learned behavior can be effectively utilized when solving downstream tasks. DISCO-DANCE outperforms other baselines in terms of average performance, and this indicates that DISCO-DANCE learned more general behaviors through guidance.

## E.3 Additional Ablation studies

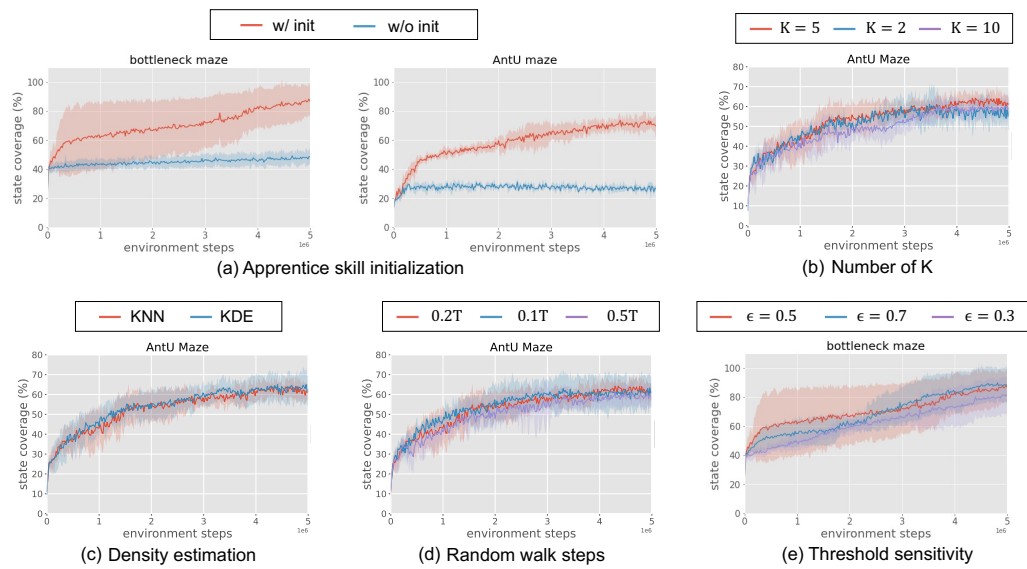

Figure 11: **Additional ablation studies.** (a) Apprentice skills initialization. (b) A number of K in K-nearest neighbors. (c) Density estimation methods. (d) A number of random walk steps. (e) Sensitivity to the accuracy threshold.

First, as we incrementally introduce new skills, we employ the guide skill to initialize the weights of the apprentice skills, allowing for direct access to unexplored states. To assess the significance of this initialization method, we compare it to a control condition in which the weights are trained from scratch. As shown in Fig 11(a), utilizing the guide skill for initialization leads to improved performance. We speculate that guide initialization directly minimizes the KL divergence with guide skill, thereby maximizing the guide reward at the onset of training.

Nest, we examined the sensitivity of the results regarding the value of k in k-nearest neighbors. As observed in our experiments (Fig. 11b), there were no significant differences in state coverage. This is because the most exploratory skill already has its terminal state far enough from the terminal states

of other skills. Therefore, adjusting the value of k higher or lower does not significantly impact the selection of guide skills. This demonstrates the robustness of DISCO-DANCE with respect to the hyperparameter k.

Additionally, to analyze how the DISCO-DANCE performs with a different measure of distance/density, we employed kernel density estimation (KDE) for selecting the guide skill. Specifically, we (i) performed random walks (as in our random walk guide selection process), (ii) trained a KDE model on the random walk arrival states, (iii) selected the state in the lowest density region among the collected random walk arrival states, and (iv) chose the skill from which that state originated as the guide skill. As shown in Fig. 11c, we found that the performance of both methods was nearly indistinguishable. This is likely due to the terminal state of the most exploratory skill being sufficiently distant from the terminal states of other skills, as was observed in the knn ablation study.

In addition, we conducted additional experiments to investigate the effect of varying the length of the random walk steps. Since the environment steps used during the random walk process are included in our total environment steps (total environment steps = general trajectory collection steps + random walk process steps), increasing the number of random walk steps leads to a decrease in the number of states utilized for model updates. As a result, we observed a slightly lower performance in the initial environment steps when comparing the result of 0.1T and 0.5T. However, we found that the models converge after 3M steps, with smaller performance differences between them.

To further examine the sensitivity of the accuracy threshold $\epsilon$, we conduct an ablation study by varying $\epsilon$ from $[0.3, 0.5, 0.7]$. As illustrated in Fig. 11e, the performance of our proposed method is found to be robust to changes in the accuracy threshold $\epsilon$.

### E.4 Comparing curriculum approach effects on other USD methods

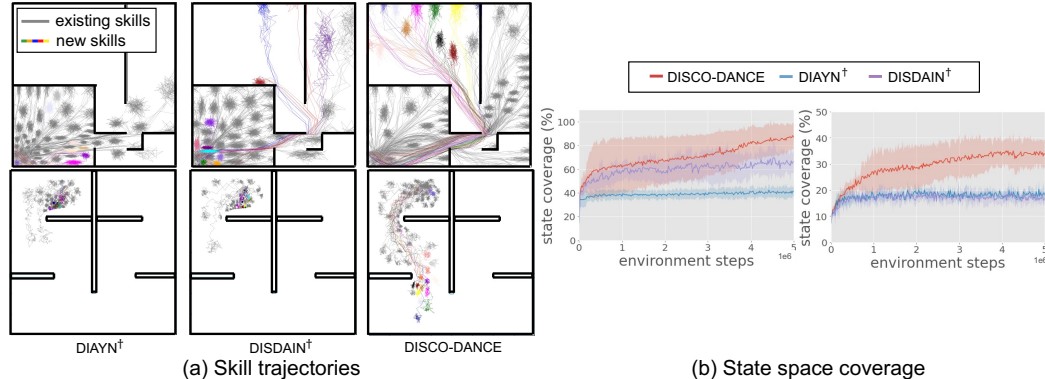

(a) Skill trajectories        (b) State space coverage

Figure 12: **Gradually increasing the number of skills for DISCO-DANCE, DIAYN, and DIS-DAIN.** (a) Qualitative results on bottleneck maze and Ant Π-maze. Colored skills indicate new skills and grey skills are previously converged skills. (b) Training curves on each environment.

To demonstrate the compatibility of our proposed method DISCO-DANCE with increasing the number of skills during training, we also incorporate a curriculum approach into the baselines which utilize discrete skill spaces. As depicted in Fig. 12a, we observe that DIAYN and DISDAIN do not benefit from curriculum learning, as the newly added skills primarily remain in proximity to the initial starting point. This results in a more congested covered state space, thus the curriculum approach fails to increase state coverage and instead decreases discriminator accuracy. In contrast, for our proposed method DISCO-DANCE, the newly added skills are able to access unexplored regions with the assistance of the guide skill.

## F    Comparison with UPSIDE

DISCO-DANCE and UPSIDE [19] have similar motivation, alleviating the inherent pessimism problem. Both of their strategies can be summarized as **"agent guides itself"** (i.e., existing skills

help new/unconverged skills to explore). In DISCO-DANCE, guide skill encourages apprentice skills (unconverged skills) to explore unexplored states by minimizing the KL divergence between guide skill and apprentice skills. While DISCO-DANCE learns a set of single skills, UPSIDE learns tree-structured policy which is composed of multiple skill segments.

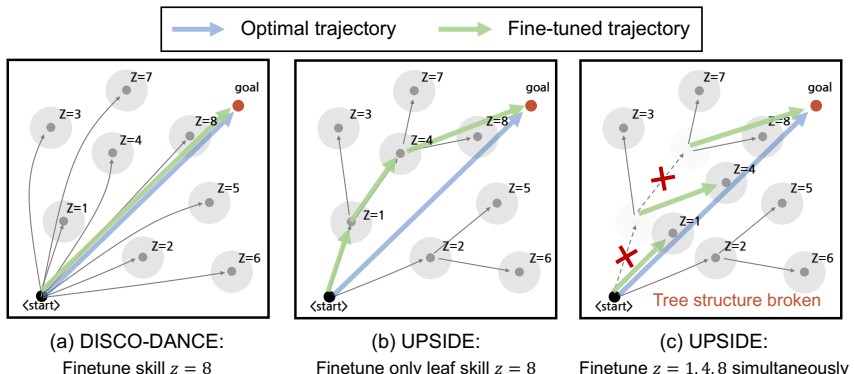

(a) DISCO-DANCE:
Finetune skill $z = 8$

(b) UPSIDE:
Finetune only leaf skill $z = 8$

(c) UPSIDE:
Finetune $z = 1, 4, 8$ simultaneously

Figure 13: **Conceptual illustration of (1) learned skills of DISCO-DANCE and UPSIDE, and (2) goal-reaching downstream task.** The black edge represents skill, and the black circle represents the terminal states of its skill. (a) After unsupervised pretraining, DISCO-DANCE obtains a set of skills that can be used individually without the need for each skill to be used together. In the finetuning stage, DISCO-DANCE selects skill $z = 8$ (i.e., skill with the highest downstream task reward) and finetunes the skill to reach the goal quickly. (b) However, to execute $z = 8$, UPSIDE requires sequential execution of all its ancestors' skills (i.e., executing $z = 1, 4, 8$ sequentially). In the finetuning stage, UPSIDE is not able to learn optimal policy since the ancestors' skills are kept fixed during finetuning. (c) If we finetune ancestors' skills simultaneously, the tree structure learned in the pretraining phase will be broken, which makes finetuning unstable and difficult.

Fig 13 illustrates an example where a total of 8 skills are learned. When learning the policy in the unsupervised pretraining stage, UPSIDE (1) selects the skill with the largest discriminator accuracy among the leaf node skills (e.g., $z = 4$), (2) adds new skills as its children nodes (e.g., $z = 7, 8$), (3) freeze the parent skills (e.g., $z = 1, 4$ are not trained) and only train newly added skills, and (4) iteratively repeat these procedures to construct the tree-structured policies (Fig 13(b)). This makes the fundamental difference between DISCO-DANCE and UPSIDE, that **UPSIDE requires sequential execution from ancestors' skills to child skill in a top-down manner** due to its tree-structured skill policies. For example, if we want to run skill $z = 8$, UPSIDE needs to execute skills sequentially ($z = 1 \rightarrow 4 \rightarrow 8$) for $T$ timesteps, where $T$ is a hyperparameter that decides how dense the generated tree will be.

One of the key objectives of unsupervised skill discovery is to utilize the learned skills as a useful primitive at the fine-tuning stage. **However, these sequential executions of UPSIDE bring significant inefficiency at the finetuning stage** due to the following reasons. Suppose the given downstream task is to reach the goal state as fast as possible, where the distance from the goal state and total execution time are given as a penalty (Fig 13). Since $z = 8$ is the skill with minimum distance from the goal, DISCO-DANCE selects skill $z = 8$ and finetunes $\pi(a|s, z = 8)$ to optimize given reward functions (Fig 13(a)). However, for UPSIDE to finetune $z = 8$, we need to finetune $z = 1, 4, 8$ simultaneously (Fig 13(c)). In that case, the tree-structured skill policies learned in the pretraining stage are broken during the finetuning stage. That is, the dictionary {"$z = 8$" : $[1, 4, 8]$} cannot be used anymore because executing $z = 1$ for $T$ timesteps does not move the agent to the original $z = 1$ terminal nodes (red $\times$ in Fig 13(c)). This means that the skill tree learned in pretraining can no longer be used, leading to inefficient finetuning.

To avoid this limitation, UPSIDE only finetuned the leaf skill in the original paper (i.e., freeze ancestors $z = 1, 4$ and only finetunes $z = 8$ in Fig 13(b)). However, this would still be ineffective because the fixed ancestors $z = 1, 4$ are not the optimal solution to solve the given downstream task (green lines from <start> to $z = 4$ nodes in Fig 13(b)). The problem becomes more serious in a long-horizon environment such as DMC (where the horizon is 1000) if we freeze all the pretrained

ancestors' skills and finetune only the last leaf skill. In contrast, DISCO-DANCE selects a single skill to fine-tune, which can be executed independently, so it does not suffer from the above problem.

There are further **differences between DISCO-DANCE and UPSIDE in the pretraining phase** in addition to the differences in the finetuning stage. In the pretraining stage, both algorithms leverage previously discovered skills for enhanced exploration. st, UPSIDE chooses one of the leaf nodes as the parent node, which additional new skill nodes will be added as children to that node (e.g., select $z = 4$ as a parent and add $z = 7, 8$ as children nodes in Fig. 13). Then $z = 7, 8$ explores the state region over the state space occupied with previous skills (e.g., $z = 1, 4$). On the other hand, DISCO-DANCE selects a guide skill from among the existing skills, and the KL divergence between the guide skill and the apprentice skill is used as a reward signal. The main difference is that **UPSIDE** selects the parent skill with the highest discriminator accuracy **(which corresponds to Fig.2b in main paper)** and **DISCO-DANCE** utilizes a proposed random walk guide selection process **(Fig.2c in main paper)**. As the most distant skill is typically the one with the highest discriminator accuracy (because the discriminator can easily classify the distant skill), the selection strategy utilized by UPSIDE is less effective in exploration, as we demonstrate in Table 2 and Fig.8 (main paper).

Lastly, both DISCO-DANCE and UPSIDE employ random walks (i.e., the diffusing component in UPSIDE) during the pretraining phase. However, **the utilization of random walks serves a completely different role in each approach**. In DISCO-DANCE, the random walk process (random walk + finding lowest density region via knn) is only utilized to select the guide skill, which is orthogonal to the policy $\pi_\theta$ learning. In contrast, *UPSIDE produces rewards from the random walks.* Specifically, UPSIDE (1) performs random walks at each skill's terminal nodes and (2) updates the policy $\pi_\theta$ and discriminator $q_\phi$ with random walked states (i.e., gets more reward if their random walked states are discriminable from other skill's random walked states). This process induces the local coverage of each skill (i.e., each skill occupies the random walked states, where other skills cannot come in).

## G    Limitations and Future Directions

DISCO-DANCE may cause cost (sample) inefficiency when measuring the density of the state distribution in environment with high dimensional input, such as control from pixel observations. In this case, we could utilize the downscaling technique (e.g., *cell representation* in Go-Explore) to reduce computational costs.

For the guide skill selection process to work stably, the termination state should be reliably reached. Therefore, when the stochasticity of the environment is too large, it would not be easy to select a guide skill only with the simple random walk process we proposed. We leave it as future work for alleviating such difficulties.

## H    Broader Impact

Unsupervised skill discovery (USD) promotes efficient learning in complex environments, allowing agents to learn without the need for explicitly labeled data. This technique has real-world applications ranging from robotic manipulation to autonomous vehicles and can also improve the AI's ability to generalize its skills to new situations. However, one of the most significant problems of USD is its sample inefficiency due to the necessity of running extensive simulations. Research like DISCO-DANCE can substantially enhance the sample efficiency of USD.

# I Visualization of learned skills

## I.1 2D mazes

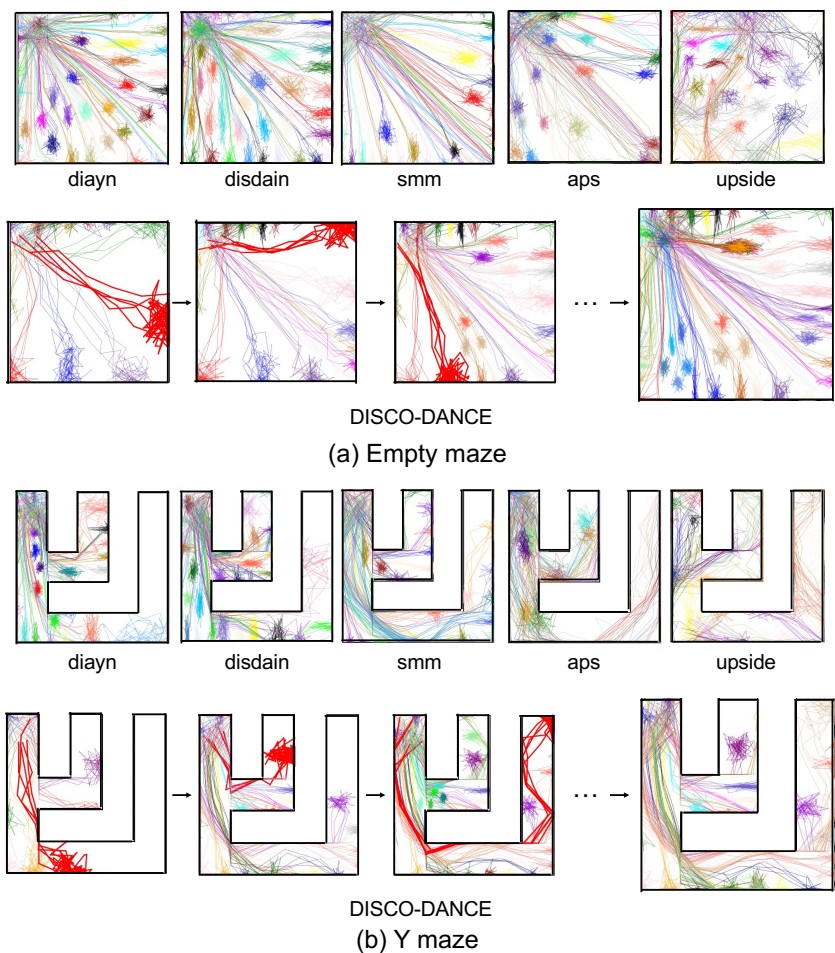

(a) Empty maze

(b) Y maze

Figure 14: **Qualitative visualization of the learned skills on Emtpy maze and Y maze.** Visualization of multiple rollouts of learned skills by baseline models. For DISCO-DANCE, we visualize our guiding procedure during training. Bold red lines indicate the guide skill.

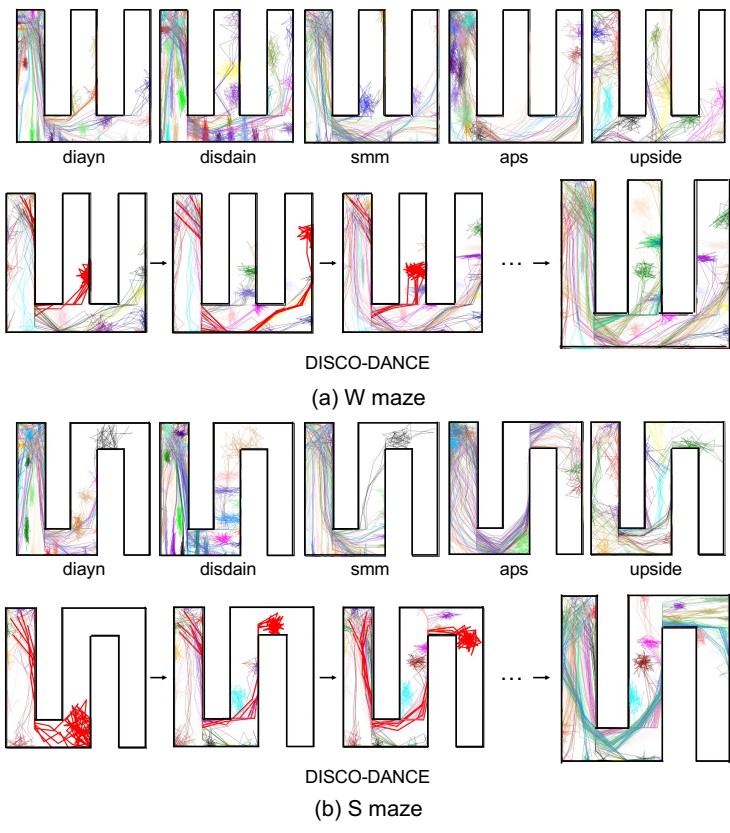

Figure 15: **Qualitative visualization of the learned skills on W maze and S maze.** Visualization of multiple rollouts of learned skills by baseline models. For DISCO-DANCE, we visualize our guiding procedure during training. Bold red lines indicate the guide skill.

## I.2 Ant Mazes

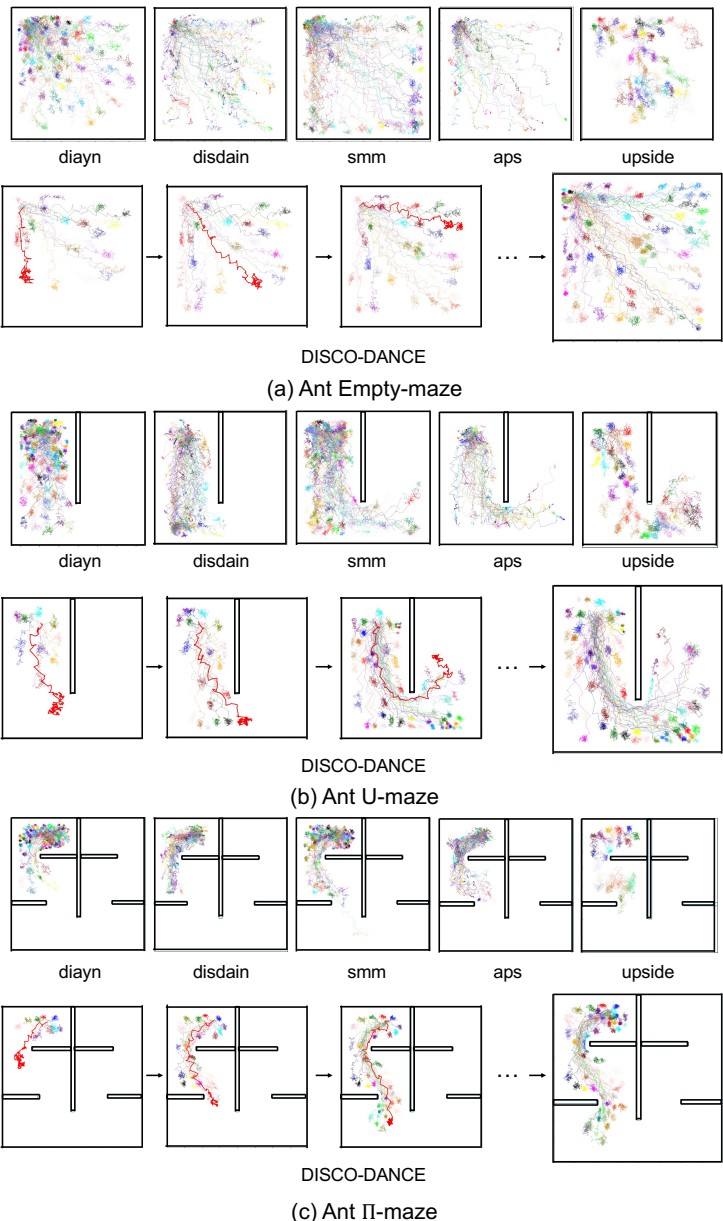

(a) Ant Empty-maze

(b) Ant U-maze

(c) Ant Π-maze

Figure 16: **Qualitative visualization of the learned skills on Ant Empty-maze, Ant U-maze, and Ant Π-maze.** Visualization of multiple rollouts of learned skills by baseline models. For DISCO-DANCE, we visualize our guiding procedure during training. Bold red lines indicate the guide skill.

## I.3 Deepmind Control Suite

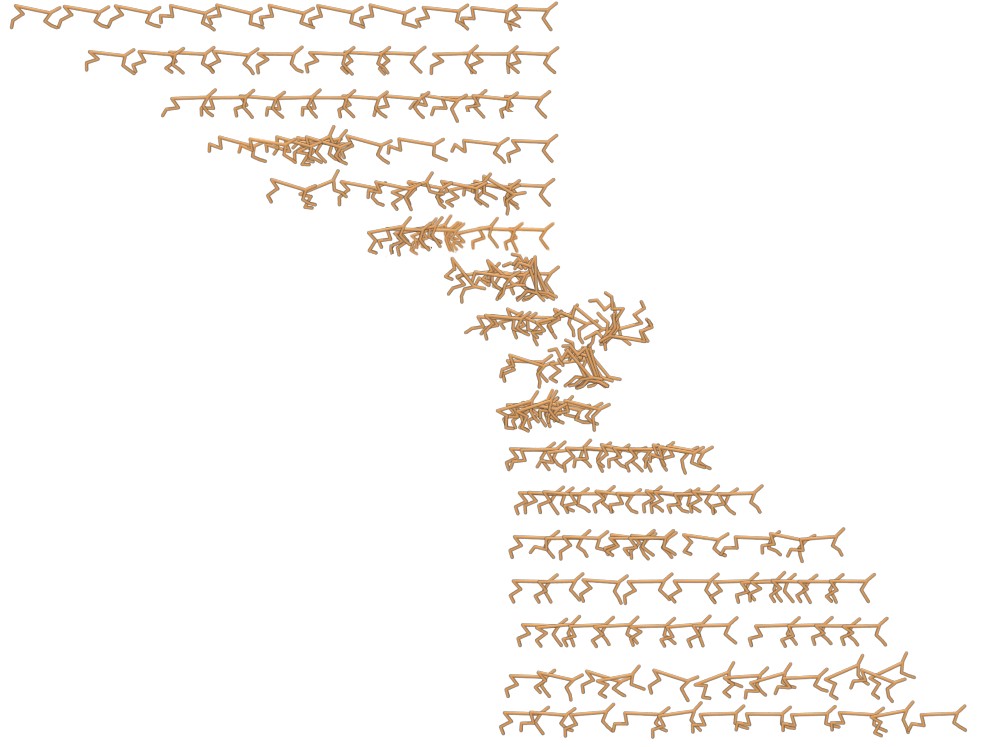

Figure 17: **Qualitative visualization of the learned skills in the cheetah in DMC.** DISCO-DANCE learned 100 skills without a reward function. In order to facilitate visualization, the skills are selected based on their corresponding terminal states.

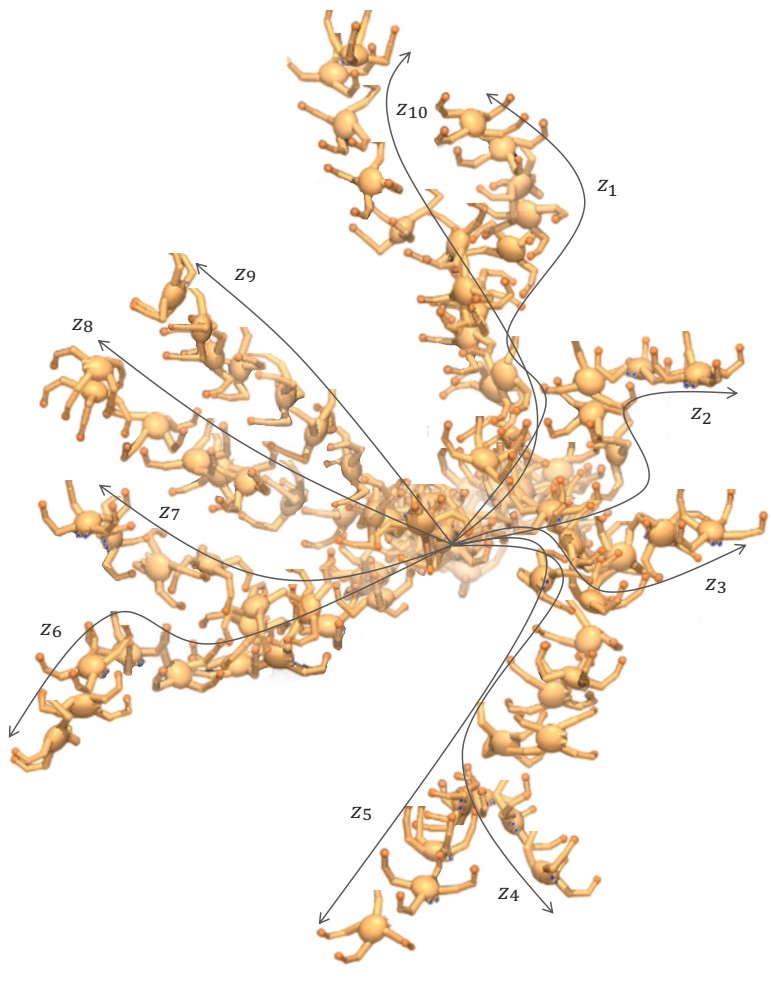

Figure 18: **Qualitative visualization of the learned skills in the quadruped in DMC.** DISCO-DANCE learned 100 skills without a reward function. In order to facilitate visualization, the skills are selected based on their corresponding terminal states.

