# OpenReview forum: "Learning to Discover Skills through Guidance"
_NeurIPS.cc/2023/Conference — NeurIPS 2023 poster_

### Official Review · Reviewer_oWju · 2023-06-14

**Soundness:** 3 good
**Presentation:** 3 good
**Contribution:** 3 good
**Rating:** 6
**Confidence:** 4

**Summary:**

This paper proposes an algorithm called DISCO-DANCE for unsupervised skill discovery in RL. The algorithm augments the mutual information reward of DIAYN with a gudiance reward. The guidance reward encourages indistinguishable or unconverged skills to follow skills that can potentially visit under-explored states, such that the over all skill collection can have broader state coverage. The authors conduct experiments on 2d navigation, Ant maze, and DMControl suites by evaluating the state coverage and the fine-tuned downstream task performance, showing that DISCO-DANCE outperforms baselines in most of these benchmarks.

**Strengths:**

+ Illustration of the main idea (i.e., find and follow a guide policy) is clear (in particular fig.1) and well-motivated.

+ The authors provide open-source code and extensive experiment details such as hyperparameters and resource requirements, which make the results presented in this paper reproducible.

+ The appendix provides extensive discussions about the algorithm's limitation and comparison with other baselines, which can help reader understand the proposed algorithm deeper.

+ Experiment results are promising.

**Weaknesses:**

+ The authors tend to address the problem unsupervised discovery in complex environments where existing methods are no longer effective, but experiments are mainly conducted on common benchmarks. I acknowledge that several maps in the navigation task are challenging, but the locomotion tasks (AntMaze and DMC) are not. Actually, baseline algorithms can outperform the proposed algorithm in terms of downstream task performance (see Appendix G).


**Questions:**

+ The authors mentioned that new skills are added if existing skills are mostly converged. In my opinion, the skill-latent-conditioned policy is an MLP with states and latents concatented as the input. If a new skill is added, the input dimension is also changed. How do you add a new skill? Do you perform a neural network surgery or specify a sufficiently large number of skills at the beginning of training?

+ Correlated with the above question, since DISCO-DANCE adds new skills during training, are the total number of skills trained by DISCO-DANCE and baselines the same in experiments?

+ In the first sentence of bstract, I think diminished bonuses (e.g. DIAYN) are not equivalent to "penalties".

+ What is the x/y-axis of fig.3(b)?

+ Fig.6 is wierd. Why not present, e.g., the success rate over 100 trials averaged over 20 seeds with error bars?

+ In Algorithm 1, the guide skill z* is not defined if the "most skills are discriminable enough" condition is not satisfied.

+ Many meaningful results are presented in the Appendix (the base RL algorithm, additional ablation studies). I think they should be linked in the main body such that readers can have a sense.

**Limitations:**

+ The authors have addressed several main limitations in appendix I.
+ The selection of the guide skill depends on the final state visited existing skills. It does not seem to be a general solution to select guide skills even for state-based environments.

---

> ### Author Rebuttal · Authors · 2023-08-10
>
> We appreciate your insightful and constructive feedback.
>
> > Q1. Why use common benchmarks (AntMaze, DMC) to evaluate the exploration ability?
>
> A1. As the reviewer pointed out, 2D mazes (especially bottleneck mazes) have more complex layouts than the other two benchmarks. However, despite this, the other two environments still require effective exploration.
>
> For Antmaze, while its environmental layout may seem simpler than a 2D maze, it is noteworthy that the dimensionality of the state space and action space is considerably higher. This intricacy makes it the agent more challenging to optimize RL policy. Consequently, Antmaze presents as a complex environment for Unsupervised Skill Discovery agents, necessitating exploration strategies. In practice, as shown in Figure 7, we can find out the agent cannot go far from the initial state as the maze layout becomes more complex (Empty-maze, U-maze, $\Pi$-maze). This empirical trend underscores the challenging nature of the Antmaze environment and affirms its suitability to measure the agent's exploration ability.
>
> We agree that DMC is not typically used as a hard exploration environment. However, for unsupervised skill discovery (USD) agents, DMC presents a non-trivial exploration challenge. For example, in Cheetah, a USD agent may easily stay near the initial location due to the pessimistic exploration problem. This occurs because the discriminator can easily distinguish skills by observing ‘slight movements’ (e.g., marginally lifting joints).  As a result, the agent may not learn to move further (i.e., run) since skills that involve staying near the starting point are already easily distinguishable. Therefore, without additional exploration signals (e.g., exploration reward), learning running skills is not easy.
>
> URLB involves various tasks within a single environment (e.g., Cheetah-run, flip). Therefore, merely excelling in one task (e.g., run) does not guarantee high performance in others (e.g., flip). The DMC benchmark evaluates the diversity of the learned skills, meaning that agents must learn a suitable set of skills for all downstream tasks in order to achieve consistently high scores across all tasks. The results indicate that DISCO-DANCE outperforms the other baseline methods, as evidenced by the aggregated IQM value (i.e., DISCO-DANCE learns diverse skills to quickly adapt to diverse tasks)
>
> > Q2. How do you add a new skill? Do you perform a neural network surgery or specify a sufficiently large number of skills at the beginning of training?
>
> A2. We set a maximum number of skills that DISCO-DANCE can acquire (e.g., 100) and initialized the network accordingly (e.g., first FC layer of policy as nn.Linear(observation_dimension + 100, hidden_dimension)).
>
> > Q3. Are the total number of skills trained by DISCO-DANCE and baselines the same in experiments?
>
> A3. Yes. We first conducted experiments with DISCO-DANCE for each environment. Based on the maximum number of skills DISCO-DANCE acquired for each environment, we set the total number of skills for other algorithms accordingly.
>
> > Q4. In the first sentence of abstract, I think diminished bonuses (e.g. DIAYN) are not equivalent to "penalties".
>
> A4. We agree. We will revise this to "a significant reduction in reward acquisition".
>
> > Q5. What is the x/y-axis of fig.3(b)?
>
> A5. The x-axis shows how much the cheetah has moved horizontally. The y-axis represents each individual skill policy that the agent has (in no particular order). We will add more explanation in the main text.
>
> > Q6. Fig.6 is weird. Why not present, e.g., the success rate over 100 trials averaged over 20 seeds with error bars?
>
> A6. Unlike typical experiments in goal-conditioned RL, where trials are performed 100 times per seed with varying goals and success rates are averaged across all seeds, our approach focused on a single, fixed goal - the most distant point from the initial state. This design choice evaluated the effectiveness of previously acquired skills in reaching challenging states (i.e., farthest state from the initial state). As a result, if a particular seed learned a policy that reaches this fixed goal, it would succeed in nearly all of its 100 trials. On the other hand, if it hasn't learned the policy of the goal yet, it would barely succeed in any of the 100 trials (i.e., the success rate for each seed is almost always either 1 or 0). Therefore, in Fig.6, instead of average success rates, we plotted the number of seeds that succeeded in reaching the goal at each timestep for simplicity.
>
> > Q7. In Algorithm 1, the guide skill z* is not defined if the "most skills are discriminable enough" condition is not satisfied.
>
> A7. Thank you for the correction. We will add "$\text{Initialize guide skill } z^* = \text{ None}$" in line 1 and
> ""$\text{If guide skill } z^* \text{ is not None:}$"" in line between 7 and 8.
>
> > Q8. Link the main paper and the Appendix.
>
> A8. Thank you for recognizing the depth of our results in the Appendix. We will add explicit references to the Appendix content in the main paper.
>
> > Q9. The selection of the guide skill depends on the final state visited existing skills. It is not a general solution to select guide skills, even for state-based environments.
>
> A9.  As outlined in Algorithm 1, line 5, the guide skill is chosen once most skills are discriminable enough (i.e., high discriminator accuracy). This indicates the skills reliably end in the same terminal state across multiple rollouts. However, as we mentioned in Appendix I, in a highly stochastic environment, it would not be straightforward to select the guide skill just by employing a random walk process. However, in a highly stochastic environment, the learned skills will visit different states for each rollout, which is a problem not only for DISCO-DANCE but for all current skill discovery algorithms. We believe this remains an important direction for future work.
>
> [1] First return, then explore. Ecoffet et al., Nature 2021.

---

> ### Author Response · Authors · 2023-08-16
> **Kind reminder for discussion**
>
> Dear Reviewer oWju (reviewer 5),
>
> We deeply appreciate the time you've dedicated to reviewing our paper.
>
> While we believe we've addressed the concerns you raised, we have yet to receive your feedback. We feel that further discussion and feedback would greatly enhance the quality of our paper. We kindly request your continued engagement and insights.
>
> Best,
> Paper 12428 authors

---

### Official Review · Reviewer_EWQN · 2023-07-06

**Soundness:** 3 good
**Presentation:** 3 good
**Contribution:** 3 good
**Rating:** 6
**Confidence:** 3

**Summary:**

This paper proposes a new unsupervised skill discovery method based on the guidance of exploration. A policy is conditioned on a latent skill variable like in prior work. The method first starts by identifying a "guide" skill variable that is likely near unexplored states, the novelty of unexplored states is measured by the density of random walk arrival states started from the terminal states of each skill. It then trains other skills that explore the vicinity of terminal states region from guide skill.


**Strengths:**

1. This paper analyzes the limitations of state coverage in previous skill discovery methods and provides nice motivation for the proposed method.

2. The empirical result in mazes and control tasks are promising.


**Weaknesses:**

1. Measuring the density of the state distribution via generating random walk arrival states from terminal states is not sample efficiency.

2. The method needs to select a guide skill that is most adjacent to the unexplored states. In the bottleneck maze tasks in Figure 3, what if all skills including the guide skill cannot pass the first room? Will this method also encourages effective exploration?


**Questions:**

1. To select guide skills for exploration, the proposed method assumes the terminal state is resettable and needs to measure the density of the state by generating about $O(PRM)$ random walks arrival states from terminal states as section 3.1 mentioned. I wonder whether is there any more sample-efficient and easy ways to do that.  e.g. Is it equivalent to picking a guide skill by directly estimating the density of terminal states sampled from history episodes or fixed-length horizon? if the skill is discriminable (MI reward is well optimized), the terminal states from the same skill may locate in a subregion of (unexplored) states.

2. What does the y-axis mean in Figure 3（b?

---

> ### Author Rebuttal · Authors · 2023-08-10
>
> We appreciate the thorough review and thoughtful comments. Please let us know if you have any further comments or feedback. We will do our best to address them.
>
> > Q1. Measuring the density of the state distribution via generating random walk arrival states from terminal states is not sample efficiency.
>
> > To select guide skills for exploration, the proposed method assumes the terminal state is resettable and needs to measure the density of the state by generating about O(PRM)  random walks arrival states from terminal states as section 3.1 mentioned. I wonder whether is there any more sample-efficient and easy ways to do that. e.g. Is it equivalent to picking a guide skill by directly estimating the density of terminal states sampled from history episodes or fixed-length horizon? if the skill is discriminable (MI reward is well optimized), the terminal states from the same skill may locate in a subregion of (unexplored) states.
>
> A1.
>
> We really appreciate your feedback. First, we wish to clarify that our approach does not strictly assume the terminal state is resettable. While leveraging the simulation environment may allow for hard resets to a terminal state of each skill, we consider such an assumption to be unrealistic. Instead, we just rollout each skill to reach the terminal state. Specifically, in the context of the Efficient Random Walk Process (detailed in Appendix F), during training, once the selected skill $z_i$ is clearly distinguishable (i.e., $q_\phi(z^i|S_T) > \epsilon$), we just perform an additional 0.2$T$ random walks from the terminal state and perform the density estimation based on these 0.2$T$ number of collected states.
>
>
> As the reviewer pointed out, there is an alternative approach where we could directly estimate density using the states from replay buffer without a random walk process.  However, such a strategy could result in a scenario as in Figure 2(b). As corroborated by the findings presented in Table 2 and Figure 8, direct estimation from the replay buffer (i.e., no random walk) during guide skill selection can potentially give rise to situations where the nearest skill to unexplored states remains unidentified. Instead, a skill distanced farthest from other skills might be chosen, consequently leading to diminished performance. This underscores that Random Walk Process is necessary for DISCO-DANCE.
>
> In addition, Random Walk Process is triggered only when the skills are sufficiently distinctive. Therefore, during training, the random walk process is not frequently executed (e.g., 8 times in 2D maze, 12 times in AntMaze, and 18 times in DMC). Moreover, for each Efficient Random Walk Process, it requires just an additional $0.2T$ steps for each skill. Considering its frequency and amount of required steps for each skill, the total steps taken by the Efficient Random Walk Process are not extensive. In summary, the Efficient Random Walk Process consumes a relatively minor portion compared to the overall training steps.
>
> It's important to note that while the random walk process does introduce some extra steps, exploration signals from the random walk process make DISCO-DANCE a more efficient explorer compared to the other baselines. Consistently, DISCO-DANCE surpasses the performance of the other baselines when measured against the same number of environment steps.
>
> > Q2. The method needs to select a guide skill that is most adjacent to the unexplored states. In the bottleneck maze tasks in Figure 3, what if all skills including the guide skill cannot pass the first room? Will this method also encourage effective exploration?
>
> A2. Yes. To demonstrate that DISCO-DANCE is beneficial in such scenarios, we provide snapshots of our 2D bottleneck maze experiments (please refer to the qualitative figure(b) in the attached pdf). As shown in the figure, even if all skills cannot pass the first room, the skill with the highest potential to approach the unexplored states (i.e., the next room) is chosen as the guide skill through the random walk process. This, in turn, motivates apprentice skills to move towards more explorable states.
>
>
>
> > Q3. What does the y-axis mean in Figure 3(b)?
>
> A4. The y-axis represents each individual skill policy that the agent has (in no particular order). We will add more explanation in the main text.

---

> ### Author Response · Authors · 2023-08-16
> **Kind reminder for discussion**
>
> Dear Reviewer EWQN (reviewer 4),
>
> We deeply appreciate the time you've dedicated to reviewing our paper.
>
> While we believe we've addressed the concerns you raised, we have yet to receive your feedback. We feel that further discussion and feedback would greatly enhance the quality of our paper. We kindly request your continued engagement and insights.
>
> Best,
> Paper 12428 authors

---

> > ### Comment · Reviewer_EWQN · 2023-08-18
> > **Response**
> >
> > Thank the authors for the clarification and additional results in Figure b. There are no major concerns for me. I increase the score to 6.

---

> > > ### Author Response · Authors · 2023-08-18
> > >
> > > We deeply appreciate your constructive feedback and updating the score. We will ensure that this clarification is adequately reflected in the revised manuscript.

---

### Official Review · Reviewer_b566 · 2023-07-06

**Soundness:** 2 fair
**Presentation:** 3 good
**Contribution:** 3 good
**Rating:** 4
**Confidence:** 4

**Summary:**

In this submission, the authors propose an unsupervised skill discovery method called DISCO-DANCE. It samples *guide skills* with random walk processes that start from the terminal states of a set of skills and use them to guide less discriminable or new skills toward those guide skills so that they can reach unexplored areas more easily. They test their method in two navigation environments (2d maze and Ant maze) and Deepmind Control Suite and compare the state space coverages and performances with the baselines.

**Strengths:**

- In terms of orignality, the main idea of this work to find a reachable state that is close to the unexplored region and to expand the skill set based on it is novel to some degree.
- The manuscript is mostly clear and easy to follow. Also, the concept figure effectively provides the intuition behind the method.
- The state space coverage problem is an important aspect of unsupervised skill discovery.

**Weaknesses:**

- The exploration issue with the mutual information (MI) objective could be more than what is described in this work. In theory, the MI objective is not supposed to contribute to the exploration meaningfully, especially in continuous control environments (Park et al. [21]), which can make this method mostly rely on the random walk processes for its exploration.
- I believe one important weakness of this submission is the random walk process. The manuscript mentions that the rise of the environmental complexity makes existing skill discovery methods less effective and motivates this work, but ironically, in complex environments (e.g., with high-dimensional state spaces), random walk would be one of the main bottlenecks in encouraging exploration. In such environments, this algorithm could require a large number of iterations.
- In terms of writing, I think it is not very fair to call the state spaces of the environments used for the benchmark *high-dimensional*. They are higher-dimensional compared to the 2D maze environment, but labeling them high-dimensional in general may not be a good standard for the field.

**Questions:**

- Do you think there could be an alternative exploration strategy other than the random walk process?

**Limitations:**

The authors state some limitations of the proposed method (difficulty in high-dimensional state spaces and stochastic environments), but I encourage the authors to consider taking the points I listed in the Weaknesses section into account.

---

> ### Author Rebuttal · Authors · 2023-08-10
>
> Thank you very much for your time and insightful comments. Please let us know if you have any further comments or feedback. We will do our best to address them.
>
> > Q1. The exploration issue with the mutual information (MI) objective could be more than what is described in this work. In theory, the MI objective is not supposed to contribute to the exploration meaningfully, especially in continuous control environments (Park et al. [21]), which can make this method mostly rely on the random walk processes for its exploration.
>
> A1. According to the findings presented in Park [21], they suggest "MI objective can be fully maximized even with small differences in states as long as different z's correspond to even marginally different ${s_{T}}'s$, not necessarily encouraging more 'interesting' skills". I believe this aligns with our discussion in lines 29-32. However, if there are any differences between the two that I might have overlooked, I would really appreciate it if the reviewer could inform us. We will reflect on it in the main paper.
>
>
> > Q2. I believe one important weakness of this submission is the random walk process. The manuscript mentions that the rise of environmental complexity makes existing skill discovery methods less effective and motivates this work, but ironically, in complex environments (e.g., with high-dimensional state spaces), random walk would be one of the main bottlenecks in encouraging exploration. In such environments, this algorithm could require a large number of iterations.
>
> A2. Please refer to general response #1, titled **General Response: Random walk process in high-dimensional state space**.
>
> > Q3. In terms of writing, I think it is unfair to call the state spaces of the environments used for the benchmark high-dimensional. They are higher-dimensional compared to the 2D maze environment, but labeling them high-dimensional in general may not be a good standard for the field.
>
> A3. We agree. We will revise it in the final manuscript.
>
> > Q4. Do you think there could be an alternative exploration strategy other than the random walk process?
>
> A4. Yes, there are alternative strategies, such as RND[1] and ICM[2], in order to choose the skill that visits the state closest to the least dense states. However, such an approach requires additional network training. Since the purpose of the guide skill selection process is merely to identify the skill with the highest potential to access unexplored states, we find that a simple random walk process is sufficient. In General response #1, we show that even in Montezuma's Revenge - which features a high dimensional pixel-based state space and is notorious for exploration challenges - 100 random walks are enough to select the guide skill.
>
>  We have shown that in our benchmarks (e.g., Ant mazes, DMC, and pixel-based Atari in General response #1), a simple random walk process successfully identifies the guide skill without additional training. We believe that in more challenging environments (e.g., real-world robotics), employing a well-designed exploration strategy will be beneficial, which we leave for future work.
>
> [1] Exploration by Random Network Distillation. Burda et al., Arxiv 2018
>
> [2] Curiosity-driven Exploration by Self-supervised Prediction. Pathak et al., ICML 2017
>
> > Q5. The authors state some limitations of the proposed method (difficulty in high-dimensional state spaces and stochastic environments), but I encourage the authors to consider taking the points I listed in the Weaknesses section into account.
>
> A5. We will add (i) an in-depth analysis of the advantage and limitations of the random walk process (including General Response #1) in Section "Limitations and Future Directions" and (ii) a more detailed explanation of the exploration issue with mutual information in Section 2.

---

> ### Author Response · Authors · 2023-08-16
> **Kind reminder for discussion**
>
> Dear Reviewer b566 (reviewer 3),
>
> We deeply appreciate the time you've dedicated to reviewing our paper.
>
> While we believe we've addressed the concerns you raised, we have yet to receive your feedback. We feel that further discussion and feedback would greatly enhance the quality of our paper. We kindly request your continued engagement and insights.
>
> Best,
> Paper 12428 authors

---

> ### Comment · Area_Chair_q19X · 2023-08-18
>
> Dear reviewer b566,
>
> you have raised issues on MI and on the random walk. May I ask you to check out the rebuttal on this (and the other reviewers) and to tell us (and the authors) what you think?
>
> Best,
> AC

---

### Official Review · Reviewer_adLT · 2023-07-06

**Soundness:** 3 good
**Presentation:** 3 good
**Contribution:** 3 good
**Rating:** 7
**Confidence:** 4

**Summary:**

This paper addresses the challenge of learning diverse skills in unsupervised reinforcement learning by introducing a method to selectively guide candidate skills to areas of the state space with low coverage from the current set of skills.  Rather than relying directly on mutual information maximization like many skill-learning approaches do, this paper proposes an algorithm for selecting guidance skills and optimizing apprentice policies through a combination of mutual information rewards and a guidance reward.  The guidance reward is high when the apprentice skill reaches similar areas of the state space as the guidance skill, which can help direct the apprentice skills to previously unexplored areas.  The paper shows that the proposed method, DISCO-DANCE, can cover more of the state space than alternate approaches by evaluating in maze navigation environments.  The paper also provides ablation studies to help understand what each part of the method contributes to the overall performance.

**Strengths:**

The paper is clearly written, with each aspect of the DISCO-DANCE algorithm explained and compared with other methods from the literature.  The contribution, both in terms of empirical results and novelty of the proposed method, is well defined.  Diagrams were effectively used to make the approach intuitive.

In addition to describing the method well, the paper evaluates the experimental performance thoroughly, comparing it to several similar methods on several domains.  The experiments use both state coverage and fine-tuning performance, and show that DISCO-DANCE performs well.

**Weaknesses:**

The main weakness of the paper mostly involves my uncertainty around the questions in the next section.  I hope that the rebuttal can help clear up confusion.

**Questions:**


1.  Appendix F describes the additional costs of the random walk process, and proposes an efficient method.  Are the additional steps used to perform random walks, even for the temporary buffer, included as environment steps in the overall training budget?  If not, this seems an unfair advantage given exclusively to DISCO-DANCE when comparing to other methods.

2.  The hyperparameters used for UPSIDE (from Appendix B) seem to limit the number of learned skills to 8 while DISCO-DANCE seems to start with more skills and has the possibility to expand.  Can you comment on this choice?  Wouldn't a higher number of skills for UPSIDE reduce the described fine-tuning difficulty by allowing more coverage?

3.  DISCO-DANCE seems to require skills to reliably end in a terminal state.  How are those states tracked and determined when the skills are still being trained?

**Limitations:**

Limitations are explicitly described and claims are not overly grand.

---

> ### Author Rebuttal · Authors · 2023-08-10
>
> We appreciate your insightful questions and positive support. Please let us know if you have any further comments or feedback. We will do our best to address them.
>
> > Q1. Appendix F describes the additional costs of the random walk process, and proposes an efficient method. Are the additional steps used to perform random walks, even for the temporary buffer, included as environment steps in the overall training budget? If not, this seems an unfair advantage given exclusively to DISCO-DANCE when comparing to other methods.
>
> A1. Yes, the additional steps used to perform random walks have been included in the overall count of environment steps for training.
>
> > Q2. The hyperparameters used for UPSIDE (from Appendix B) seem to limit the number of learned skills to 8 while DISCO-DANCE seems to start with more skills and has the possibility to expand. Can you comment on this choice? Wouldn't a higher number of skills for UPSIDE reduce the described fine-tuning difficulty by allowing more coverage?
>
> A2. UPSIDE sets the initial state as the parent node, from which it generates $N_\text{start}=2$ skills (i.e., leaf nodes) that move a short distance from parent node. UPSIDE then incrementally adds new leaf node to sufficiently cover the state space around the parent node, up to a maximum of $N_\text{max}=8$ nodes. Subsequently, among these leaf nodes, the most discriminable skill is chosen and set as a new parent node, which then generates its leaf nodes. This procedure of skill tree expansion is iteratively executed. Hence, if there is remaining state space in the environment, UPSIDE can continue to increase the number of skills through the aforementioned process until the end of training.
>
> In our experiments, we performed hyper-parameter search for $N_\text{max}$ from 6 to 10, and found that there was minimal difference in performance. Therefore, we selected 8 for $N_\text{max}$ (which was used in orignal paper).
>
>
> > Q3. DISCO-DANCE seems to require skills to reliably end in a terminal state. How are those states tracked and determined when the skills are still being trained?
>
> A3. As outlined in Algorithm 1 on page 5, line 5, the guide skill is chosen once the majority of skills are discriminable enough (i.e., high discriminator accuracy). This indicates that the terminal states achieved by each skill are consistent. This ensures that, when selecting the guide skill, most skills will consistently end in the (almost) same terminal state across multiple rollouts.

---

> > ### Comment · Reviewer_adLT · 2023-08-16
> >
> > Thank you for your response, your clarifications around UPSIDE are especially helpful.  I have no further questions.

---

> > > ### Author Response · Authors · 2023-08-17
> > >
> > > Thank you for the constructive feedback and positive evaluation. We will provide more details about UPSIDE in the Appendix.

---

> ### Author Response · Authors · 2023-08-16
> **Kind reminder for discussion**
>
> Dear Reviewer adLT (reviewer. 2),
>
> We deeply appreciate the time you've dedicated to reviewing our paper.
>
> While we believe we've addressed the concerns you raised, we have yet to receive your feedback. We feel that further discussion and feedback would greatly enhance the quality of our paper. We kindly request your continued engagement and insights.
>
> Best,
> Paper 12428 authors

---

### Official Review · Reviewer_gakY · 2023-07-08

**Soundness:** 3 good
**Presentation:** 3 good
**Contribution:** 1 poor
**Rating:** 3
**Confidence:** 5

**Summary:**

The paper introduce a reward shaping approach called DISCO-DANCE to enhance exploration in unexplored states in the context of latent variable skill learning in self-supervised RL context. The approach consist in defining a guide skill with the highest potential for reaching unexplored states for a given environment, then select unconverged skills and incentivize them to follow the guide skill expressed at terminal state, helping them bypass state regions with low rewards. Finally, the approach disperse the skills to maximize their distinctiveness, resulting in a set of skills that cover a wide range of states. DISCO-DANCE fills the pathway to unexplored regions with positive rewards. The approach is compared in navigation and locomotion scenarios surpassing some previous methods in terms of exploring the state space and performing well in navigation tasks in 2D mazes and Ant mazes.

**Strengths:**

The paper is well written and explain rather well his approach.
The problem of skill learning with self-supervised learning is important and actual.
The justification is reasonably clear, maybe it would have been interesting to discuss more the difference between environment were most degrees of freedom are part of the action space, like in locomotion and the cases of manipulation that poses the most issues.
The experiment use classic but simple 2D navigation and simulated locomotion scenarios to illustrates the benefit.

**Weaknesses:**

The comparison is rather limited, we woud have like to see DADS, MUSIC and LSD for example.
We would also have like to see experiments in more known challenging environment like manipulation where MI is the most in trouble.

**Questions:**

Why not having compare with DADS, MUSIC and LSD?
Is the random walk used at the last step of the guide skill discovery is scalable to larger state environments ?

**Limitations:**

The experiments are rather limited to justify the benefit of the approach.
The random walk as last step of the definition of the guide skill definition isn't shown to be scalable to larger state environments.

---

> ### Author Rebuttal · Authors · 2023-08-10
>
> We appreciate your valuable feedback. Please let us know if you have any further comments or feedback. We will do our best to address them.
>
> > Q1. Why not having compare with DADS, MUSIC and LSD? We would also have like to see experiments in more known challenging environment like manipulation where MI is the most in trouble.
>
> A1. We agree that including more baseline algorithms such as DADS, MUSIC, and LSD [1,2,3] would further increase the impact of DISCO-DANCE, and it's indeed a valuable direction for subsequent work. Our primary focus in this work, however, is to tackle an issue of pessimistic exploration inherent in MI-based skill discovery algorithms. As highlighted in Section 5 (line 297:302, 312:315), our chosen baselines are algorithms which focus on devising an effective auxiliary reward  ($r_\text{exploration}$) to help exploration. On the other hand, DADS, MUSIC, and LSD mainly focus on modeling the representation of skills ($r_\text{skill}$). This is why these algorithms are not included in our paper as baselines.
>
> Furthermore, our experimental environments are chosen to measure the effectiveness of the devised $r_\text{exploration}$. To evaluate how well $r_\text{exploration}$ helps overcome pessimistic exploration, we utilized environments such as 2D and Ant mazes. In such settings, our approach outperformed baseline algorithms, validating the effectiveness of $r_\text{exploration}^\text{DISCO-DANCE}$ in enhancing the exploration. Moreover, to demonstrate DISCO-DANCE aids in not only navigation tasks but also in learning useful skills in more diverse settings, we conducted experiments on a widely-used continuous control benchmark, URLB [4], and were also able to surpass all regarding baselines.
>
> To clarify the significance of devising a well-designed $r_\text{exploration}$, we conduct additional experiments on LSD (with discrete skills) within a 2D bottleneck maze (please refer to Figure \(c\) in the attached pdf). Utilizing their open-source code, we optimized three important hyperparameters (learning rate, entropy coefficient of SAC ($\alpha$), dimension of skills). As the main objective of LSD is to obtain 'far-reaching' skills, far-reaching skills are well learned in an obstacle-free empty environment (similar to Figure 4 in the LSD paper).
>
> However, Figure \(c\) shows that LSD struggles to efficiently navigate in bottleneck maze, underscoring that replacing $r_\text{skill}^\text{DIAYN}$ with $r_\text{skill}^\text{LSD}$ is not enough for solving hard-exploration environments. This emphasizes the importance of well-designed $r_\text{exploration}$, regardless of what $r_\text{skill}$ is used.
>
> [1] Dynamics-Aware Unsupervised Discovery of Skills. Sharma et al., ICLR 2020
>
> [2] Mutual Information State Intrinsic Control. Zhao et al., ICLR 2021
>
> [3] Lipchitz-constrained Unsupervised Skill Discovery. Part et al., ICLR 2022
>
> [4] URLB: Unsupervised Reinforcement Learning Benchmark. Laskin et al., NeurIPS 2021
>
>
> > Q2. Is the random walk used at the last step of the guide skill discovery is scalable to larger state environments?
>
> A2. Please refer to General Response #1, titled **General Response: Random walk process in high-dimensional state space**.

---

> ### Author Response · Authors · 2023-08-16
> **Kind reminder for discussion**
>
> Dear Reviewer gakY (reviewer 1),
>
> We deeply appreciate the time you've dedicated to reviewing our paper.
>
> While we believe we've addressed the concerns you raised, we have yet to receive your feedback. We feel that further discussion and feedback would greatly enhance the quality of our paper. We kindly request your continued engagement and insights.
>
> Best,
> Paper 12428 authors

---

> ### Comment · Area_Chair_q19X · 2023-08-18
>
> Dear reviewer gakY,
>
> you raised concerns about comparisons with baselines DADS, MUSIC and LSD. The authors have responded to this first by explaining their choice of comparisons, and also by providing experiments with LSD.
>
> May I ask you to acknowledge the rebuttal and to tell us (and the authors) what you think?
>
> Thanks,
> AC

---

> ### Author Response · Authors · 2023-08-18
>
> Dear Reviewer gakY,
>
> Regarding the comparison with LSD, in addition to our prior LSD experiments on 2D bottleneck maze, we are in the process of organizing an extra experiment on the Ant-$\Pi$ maze to demonstrate the robustness of $r_\text{exploration}^\text{DISCO-DANCE}$. We hope this additional experiment will address the Reviewer gakY's concerns well. We intend to share the outcomes by Aug 20th 2am EDT.
>
> Best,
> Paper 12428 authors

---

> ### Author Response · Authors · 2023-08-20
> **Comparative Analysis of DISCO-DANCE and LSD**
>
> Dear Reviewer gakY,
>
> We appreciate your feedback regarding the need for comparisons with algorithms such as DADS, MUSIC, and LSD. Initially, as discussed in our response, we classified skill discovery methods into two broad categories:
>
> 1. Algorithms that utilize $r_{skill}$ with a mutual information objective while focusing on constructing an effective auxiliary exploration reward $r_{exploration}$.
> 2. Methods that seek to redefine $r_{skill}$ beyond mutual information.
>
> In our earlier communication, we positioned DISCO-DANCE in the former category, believing that its closest counterparts were the algorithms targeting the exploration reward. This informed our initial decision to exclude DADS, MUSIC, and LSD from our comparisons, as they were primarily aligned with the latter category.
>
> However, upon your feedback and deeper reflection, we recognize the value of broadening our comparative analysis. Even if DADS, MUSIC, and LSD primarily fall into the second category, these algorithms may overcome the inherent pessimism associated with the mutual information objective. In response, we have decided to extend our comparative analysis with LSD to ensure a comprehensive evaluation of skill discovery algorithms.
>
> Among those, we selected LSD, the algorithm known as the top performer among the reviewer has suggested. Due to time constraints, we narrowed our focus to the Ant-$\Pi$ maze, the environment where the agent suffer from exhaustive exploration.
>
> | $r_\text{skill}$ | $r_\text{exploration}$ |state coverage |
> | -------- | -------- | -------- |
> | DIAYN     | None     | 22.50±3.34     |
> | DIAYN     | DISCO-DANCE     | 39.00±4.85     |
> | LSD     | None     | 38.80±3.34     |
> | LSD     | DISCO-DANCE     | 45.80±3.34     |
>
> The above table illustrates the performance comparison with LSD. A noteworthy observation is that, unlike in the 2D-bottleneck-maze (Fig.c in attached pdf), both DISCO-DANCE and LSD exhibit comparable performance in the Ant-$\Pi$ maze. This potentially underscores LSD's capability to mitigate the inherent pessimism of mutual information objective.
>
> While the performance of LSD is noteworthy, it doesn't diminish the significance of DISCO-DANCE. This is because $r_{skill}$, when decoupled from the mutual information objective, and $r_{exploration}$ as guided by DISCO-DANCE, can work in tandem, complement each other. We tested this by combining DISCO-DANCE with LSD, which led to significant performance improvements. Thus, in environments that are challenging to explore, DISCO-DANCE can serve as a key role in improving exploration.
>
> In summary, we acknowledge that our initial experiments primarily utilized MI-based algorithms, specifically DIAYN.  Yet, as shown in our additional experiments, DISCO-DANCE not only outperforms LSD in both the 2D-bottleneck-maze and Ant-$\Pi$ maze, but also exhibits augmented performance improvements when integrated with LSD. We are also in the process of conducting experiments with MUSIC to provide a more comprehensive evaluation across various environments. In our revised manuscript, we will incorporate these results, along with additional ablation studies where DISCO-DANCE is utilized on top of different skill discovery objectives.
>
> We hope our response alleviates your concerns.
>
> Best, Paper 12428 authors

---

### Author Rebuttal · Authors · 2023-08-10

**Response to all reviewers**

 We deeply appreciate all five reviewers for their thoughtful feedback and valuable suggestions. R1, R2, R3, R4 and R5 indicate reviewer gakY, reviewer adLT, reviewer b566, reviewer EWQN, reviewer oWju respectively.

 Reviewers identified the following strength in our submission:

- The main idea is intuitive and well-motivated (R2, R3, R4, R5).
- The proposed method was evaluated against several methods in various domains, and the empirical results are promising (R2, R4, R5).

 At the same time, reviewers identified the following weaknesses in our submission:

- Scalability of Random Walk Process to high dimensional state spaces (R1, R3).
- Missing baselines and missing experiments in manipulation environments (R1).
- Sample inefficiency of Random Walk Process (R4).


We hope our responses address all reviewer’s concerns, and we welcome any additional comments and clarifications.


**General Response: Random walk process in high-dimensional state space**

Reviewer R1 and R3 raised questions regarding the potential scalability of our random walk process within high-dimensional state spaces. To address this, we've identified two essential questions to consider:

1. When the agent starts to explore from an arbitrary terminal state, can the agent visit a diverse range of states through the random walk process?
2. Given the diverse range of visited states, is it possible to identify the terminal state (guide skill) within the least explored region?


In response, we conducted a synthetic experiment on Montezuma's Revenge, which is a high-dimensional pixel-based environment characterized by an 84×84×3 input and is well-known for its exploration challenges.

Figure (a) in the attached PDF provides a visual illustration of our random walk approach within Montezuma's Revenge. In this experiment, we:

1. Randomly reset the agent in the inital room of Montezuma's Revenge.
2. Execute a random walk for 100 steps from the randomly initialized state.
3. Iterate the step 1,2 for N cycles.


This experimental design mirrors the algorithmic design of DISCO-DANCE where each reinitialized point indicates the terminal state of a different skill. Correspondingly, each skill undergoes a random walk for 100 steps, aligning with the parameters used for our paper (i.e., P=N, R=100, M=1 in Equation 4).

> 1. When the agent starts to explore from an arbitrary terminal state, can the agent visit a diverse range of states through the random walk process?

As illustrated by Figure (a), even in such a high-dimensional state space, the agent was able to visit a variety of states within just 100 random steps. For instance, for Skill 'I', the agent was able to move to another room. And with for Skill 'J', the agent successfully picked up the key. This experimental result supports the versatility of the random walk process, even when the environment is high-dimensional.


> 2. Given the diverse range of visited states, is it possible to identify the terminal state (guide skill) within the least explored region?


After random walk, our aim shifts to identifying the guide skill within the least explored area. For density estimation, we employed the cell-centric technique from Go-Explore [1]: (i) segmenting the aggregated states into discrete cells, (ii) count the number of each cell's visitation, and (iii) select the least visited cell. As emphasized in Figure (a), the cell marked in red is selected as guide skill, indicating that skill 'I' is the prime candidate for guide skill in DISCO-DANCE.


This combined approach of exploration through random walk then identifying unique states (which is used in DISCO-DANCE), parallels the approach adopted by Go-Explore. This technique has consistently demonstrated its efficacy across varied domains, including Atari games and robotic settings.

In summary, we believe our synthetic experiment affirms the scalability of our random walk process in DISCO-DANCE, even within a high-dimensional pixel-based environment. In the revised manuscript, we'll be integrating these insights into the 'Limitations and Future Directions' section.

[1] First return, then explore. Ecoffet et al., Nature 2021.

---

### Decision · Program_Chairs · 2023-09-21

**Decision:**

Accept (poster)

**Comment:**

The paper describes a method for unsupervised skill discovery based on a guidance principle, where a random walk process guides the discovery process towards new unexplored areas in the state space. Initial ratings were mixed (5,4,6,3,7) and evolved to (6,4,6,3,7). Reviewers appreciated an interesting idea (random walks to discover unexplored areas), a well written paper,

Weaknesses raised were missing baselines, links to mutual information, sample efficiency, difficulty of the tasks (ab application to manipulation was requested).

The AC's own reading confirmed the strengths of the paper and some of weaknesses raised by the reviewers, but the AC did not side with all of them. In particular, the AC considered that the novelty and interestingness of the paper (guidance through random walk) outweights the issues, which are not important enough to prevent the paper from having a positive impact on the community. The AC recommends acceptance.

This paper was discussed between the AC and SAC.